# Nature exposure induces analgesic effects by acting on nociception-related neural processing

Maximilian O. Steininger [1], Mathew P. White [2,3,4], Lukas Lengersdorff[1], Lei Zhang [1,5,6,7], Alexander J. Smalley [3], Simone Kühn[8,9] & Claus Lamm [1,2,4] ✉

Nature exposure has numerous health benefits and might reduce self-reported acute pain. Given the multi-faceted and subjective quality of pain and methodological limitations of prior research, it is unclear whether the evidence indicates genuine analgesic effects or results from domain-general effects and subjective reporting biases. This preregistered neuroimaging study investigates how nature modulates nociception-related and domain-general brain responses to acute pain. Healthy participants (N = 49) receiving electrical shocks report lower pain when exposed to virtual nature compared to matched urban or indoor control settings. Multi-voxel signatures of pain-related brain activation patterns demonstrate that this subjective analgesic effect is associated with reductions in nociception-related rather than domain-general cognitive-emotional neural pain processing. Preregistered region-of-interest analyses corroborate these results, highlighting reduced activation of areas connected to somatosensory aspects of pain processing (thalamus, secondary somatosensory cortex, and posterior insula). These findings demonstrate that virtual nature exposure enables genuine analgesic effects through changes in nociceptive and somatosensory processing, advancing our understanding of how nature may be used to complement non-pharmacological pain treatment. That this analgesic effect can be achieved with easy-to-administer virtual nature exposure has important practical implications and opens novel avenues for research on the precise mechanisms by which nature impacts our mind and brain.

Natural settings such as parks, woodlands, coastlines, and their constituent elements, including plants, sunsets, and natural soundscapes, can protect and promote a range of health and well-being outcomes[1–3]. People who live in greener neighborhoods tend to react less strongly to stressors[4] and have better mental health in the long term[5], regular nature visitors report fewer negative and higher positive emotional states[6], and even short experimental nature exposures can positively impact subjective and neural indicators of well-being[7]. Theories connecting nature

[1]Social, Cognitive, and Affective Neuroscience Unit, Department of Cognition, Emotion, and Methods in Psychology, Faculty of Psychology, University of Vienna, Vienna, Austria. [2]Cognitive Science Hub, University of Vienna, Vienna, Austria. [3]European Centre for Environment and Human Health, University of Exeter, Truro, UK. [4]Environment and Climate Research Hub, University of Vienna, Vienna, Austria. [5]Centre for Human Brain Health, School of Psychology, University of Birmingham, Birmingham, UK. [6]Institute for Mental Health, School of Psychology, University of Birmingham, Birmingham, UK. [7]Centre for Developmental Science, School of Psychology, University of Birmingham, Birmingham, UK. [8]Center for Environmental Neuroscience, Max Planck Institute for Human Development, Berlin, Germany. [9]Department of Psychiatry, University Medical Center Hamburg-Eppendorf, Hamburg, Germany. ✉e-mail: claus.lamm@univie.ac.at

and health underscore various aspects that render certain natural environments particularly salutary. While stress recovery theory (SRT) proposes that the presence of natural, non-threatening content elicits positive affective responses and aids recovery from stress[8], attention restoration theory (ART) puts a stronger emphasis on nature's ability to replenish voluntary attentional resources[9]. According to ART, certain natural settings encompass numerous elements that captivate human attention in a unique and effortless way. While differing in focus, both theories highlight nature's capacity to benefit human health, an assumption that has been substantiated by a multitude of evidence.

Of particular relevance to this study, natural settings may even have the potential to reduce acute pain[10–12]. Forty years ago, Ulrich (1984) showed that patients recovering from surgery were given fewer analgesics to manage pain, had more positive healthcare provider notes, and left the hospital earlier when having a window view of trees compared to a brick wall[11]. Similar results have subsequently been reported using various forms of nature exposure during diverse pain-related settings (e.g., invasive medical procedures such as dental treatments or bronchoscopy[10,12]). However, the evidence to date has several limitations.

For instance, due to a lack of proper experimental controls previous work has been unable to fully assess whether it is nature specifically that reduces pain. Most studies have either not compared nature exposure to an alternative stimulation or used control conditions that were not carefully matched on key aspects such as low- or high-level visual features or subjective beauty[13,14]. For example, nature is often juxtaposed with aesthetically unpleasing or stressful settings, such as unappealing and busy urban environments. It thus remains unclear whether natural scenes reduce pain or if the alternative environments exacerbate it through their negative characteristics[8]. Carefully controlled experimental designs are required to assess this conclusively, ensuring that nature and control stimulations are closely matched on relevant key features.

Furthermore, most prior research has relied on self-report measures of pain, which, whilst important, are limited in two central regards. First, self-reports make it challenging to capture the multifaceted quality of pain. Pain entails several components, ranging from lower-level sensory aspects, such as nociception and its neural processing, to higher-level components, involving affective, cognitive, and motivational processes and their associated neural responses[15]. The sensory aspects reflect people's ability to identify from where in the body a painful stimulus originated, how intense it is, and what type of pain is perceived. The cognitive-affective and motivational aspects entail feelings of unpleasantness towards the stimulus and the inclination to engage in protective behavior, as well as pain-related affect regulation. Although separate ratings of pain intensity and unpleasantness might experimentally disentangle these aspects at a subjective level[16], such self-reports are susceptible to various confounding influences[17]. Second, affective, cognitive, and motivational processes associated with pain also play a role in other types of subjective experiences and thus may not entirely reflect pain-specific but rather domain-general pain-related processing[18]. We thus cannot exclude that previous findings were primarily driven by the effects of nature on such domain-general processes and, therefore, lack specificity for pain. Moreover, self-report is limited by individual constraints in self-perception and meta-cognition, and beliefs about how nature exposure will influence one's pain sensitivity, alongside other types of experimental demand effects that may have unintentionally influenced prior findings[19].

Neuroimaging techniques have been suggested as a possible way to complement self-report and facilitate a systems-level approach to the brain bases of pain. Indeed, experiencing pain involves numerous interconnected brain structures, and particular brain regions may be associated with distinct pain components[20]. For example, while the posterior insula (pINS) and the secondary somatosensory cortex (S2)

are predominantly involved in early 'lower-level' nociception-related processing, 'higher-level' components incorporating sensory, emotional and motivational aspects are associated with regions such as the anterior midcingulate (aMCC) and the prefrontal (PFC) cortex[20,21]. Distinguishing neural processes as pertaining to 'lower-level' vs. 'higher-level' is a useful yet necessarily imprecise heuristic (which is why we put them under quotation marks here). Although individual brain regions rarely serve a singular role in pain processing, typically integrating both lower- and higher-level pain components[22], evidence indicates that certain regions specialize in one aspect of pain over the other[23–26]. For instance, the pINS responds to noxious stimuli with minimal latency, is highly interconnected with sensorimotor cortices, and primarily encodes both the intensity and bodily location of a stimulus. In contrast, the anterior insula (aINS) exhibits a delayed response, is more strongly connected with prefrontal regions, and integrates information from the pINS to generate an emotional response to pain[23,26]. While acknowledging the complexity of pain processing and recognizing that the activation of brain areas aligns more with a gradient than a supposed dichotomy of underlying computations, evaluating brain responses during acute pain could yield more refined and less subjective assessments of the various processes underpinning the multifaceted quality of pain and help to disentangle if lower- or higher-level processes are impacted.

In this respect, recent advancements in pain research are of particular value. For example, machine learning approaches and multivariate brain patterns have been applied to neuroimaging data to identify and differentiate between various aspects of pain with even higher precision and validity when compared to the analysis of single isolated brain regions[27]. Specifically, two prominent multivoxel patterns, the neurologic pain signature (NPS[27]) and the stimulus intensity independent pain signature-1 (SIIPS1[28]) have been developed to investigate and differentiate between lower-level and higher-level pain-related processing, respectively. The NPS tracks the intensity of a painful stimulus and involves brain regions that receive nociceptive afferents[29], thus capturing processes connected to nociception and lower-level sensory processing. The SIIPS1 has been developed to assess pain-related brain activity beyond nociception and captures aspects such as motivational value and emotional or cognitive context[28]. Importantly, the NPS has been shown to predict pain individually with high sensitivity and specificity, allowing the disambiguation from domain-general and non-specific processes such as negative emotion or cognitive appraisal, which also play a role in pain processing[27]. In contrast, the SIIPS1 was explicitly developed to capture variance in pain after accounting for sensory processing. It primarily includes brain regions linked to cognitive and affective functions, mediates expectancy effects on pain, and may be shaped by broader cognitive and affective processes that are not exclusively tied to pain[28,30]. Thus, it rather captures the engagement of domain-general processes while people experience acute pain. The aim of our study was to exploit these recent methodological developments and neuroscientific insights to better understand the neural processes and mechanisms by which nature exposure might lead to the reduction of painful experiences. Besides advancing our basic knowledge, such research may have considerable importance for efforts to complement pharmaceutical treatment approaches, with their well-documented negative side effects and addictive properties[31].

To address these research gaps, we conducted a preregistered repeat-crossover functional magnetic resonance imaging (fMRI) experiment. In the fMRI scanner, healthy human participants were exposed to carefully matched virtual natural and urban scenes, as well as an indoor setting control condition, while experiencing electric shocks that induced individually calibrated acute transient pain (Fig. 1). Combining multivoxel brain signature approaches (both NPS and SIIPS1) with analyses of distinct pain-responsive brain areas allowed us

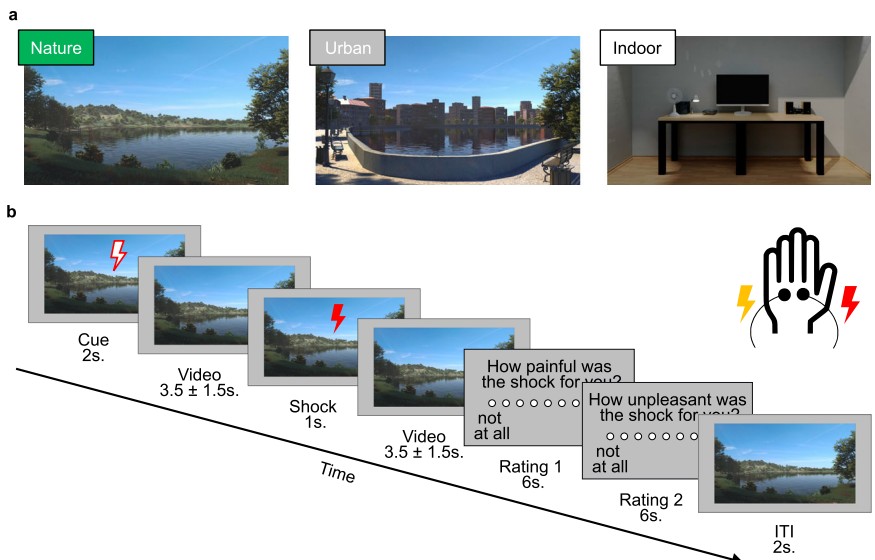

**Fig. 1 | Stimuli and trial structure of the experiment. a** Stimuli depicting a natural, an urban, and an indoor environment. A matching soundscape accompanied each visual stimulus. The three pain runs had a total duration of 9 min each, during which one environment was accompanied by 16 painful and 16 nonpainful shocks. All participants were exposed to all environments (in counterbalanced order). **b** Structure and timeline of an example trial. First, a cue indicating the intensity of the next shock (red = painful, yellow = not painful) was presented for 2000 milliseconds (ms). Second, a variable interval of 3500 ± 1500 ms was shown.

Third, a cue indicating the intensity of the shock was presented for 1000 ms, accompanied by an electrical shock with a duration of 500 ms. Fourth, a variable interval of 3500 ± 1500 ms followed. Fifth, after each third trial, participants rated the shock's intensity and unpleasantness at 6000 ms each. Sixth, each trial ended with an intertrial interval (ITI) presented for 2000 ms. The environmental stimulus was presented simultaneously except for the rating phase during each trial. Electrical painful and non-painful shocks were administered to the dorsum of the left hand with a separate electrode.

to explore the impact of nature stimuli (vs. urban and indoor controls) on different aspects of the pain-processing hierarchy.

Based on previous research, yet using a carefully designed experiment with highly controlled experimental stimuli (Methods for details), we hypothesized that exposure to nature compared to urban or indoor control settings would reduce self-reported pain. For the neuroimaging data, with which we aimed to significantly extend previous behavioral research, we predicted that pain-related neural activity would be reduced by exposure to nature compared to the control conditions. Both hypotheses were preregistered. While we expected reductions in brain responses associated with lower-level nociception-related or higher-level pain-related emotional-cognitive processes, the lack of prior neuroimaging research precluded specific predictions about which of the two processes would be impacted preferentially.

## Results

### Nature stimuli reduce self-reported pain

We used immediate self-report ratings of experienced pain intensity and unpleasantness to study participants' subjective pain responses. With the intensity ratings, we intended to capture the sensory-discriminative, and thus nociception-related, aspects of pain, while the unpleasantness ratings aimed to measure higher-level cognitive-emotional and motivational features[16,21]. Participants were carefully instructed to discriminate both aspects and rated each separately on a scale from zero ("not at all painful/unpleasant") to eight ("very painful/unpleasant"; see Experimental Procedures). Statistical inferences of the self-report data were based on linear mixed modeling (LMM; see Methods and Supplementary Information).

Supporting our preregistered hypothesis, we found a significant main effect of environment (nature, urban, or indoor) on the immediate ratings [i.e., pooled intensity and unpleasantness ratings, $F_{(2,48.14)} = 12.49$, $p < 0.001$]. Planned pairwise contrasts revealed that self-reported pain was lower in the nature vs. urban [$b = -0.54$, SE = 0.12, $t_{(48)} = -4.46$, $p < 0.001$ one-tailed, 95% CI = [−0.782, −0.296],

$d_{rm} = -0.53$] and indoor condition [$b = -0.48$, SE = 0.11, $t_{(48)} = -4.14$, $p < 0.001$ one-tailed, 95% CI = [−0.708, −0.245], $d_{rm} = -0.44$], with urban and indoor conditions not differing [$b = 0.06$, SE = 0.12, $t_{(48)} = 0.52$, $p = 0.60$, 95% CI = [−0.178, 0.303], $d_{rm} = 0.06$]. We found a significant interaction effect of environment*rating type [$F_{(2,81.11)} = 9.19$, $p < 0.001$)]. Investigations of the beta parameters and planned pairwise contrasts suggested that while both types of ratings were lower in the nature environment compared to the urban or indoor settings, the magnitude of change differed between unpleasantness and intensity ratings. As displayed in Fig. 2a, b, the differences between nature and the other two conditions were larger for the unpleasantness than for the intensity ratings, with effect sizes representing medium and small magnitudes, respectively. Specifically, planned pairwise contrasts revealed a significant difference in intensity ratings between nature vs. urban [$b = -0.25$, SE = 0.12, $t_{(48)} = -2.14$, $p = 0.018$ one-tailed, 95% CI = [−0.482, −0.015], $d_{rm} = -0.26$] and nature vs. indoor [$b = -0.29$, SE = 0.11, $t_{(48)} = -2.67$, $p = 0.005$ one-tailed, 95% CI = −0.514, −0.073], $d_{rm} = -0.31$] but not for urban vs. indoor [$b = -0.05$, SE = 0.12, $t_{(48)} = -0.38$, $p = 0.71$, 95% CI = [−0.283, 0.193], $d_{rm} = -0.05$]. Similarly, the unpleasantness ratings showed a significant difference comparing nature vs. urban [$b = -0.83$, SE = 0.16, $t_{(48)} = -5.23$, $p < 0.001$ one-tailed, 95% CI = [−1.149, −0.511], $d_{rm} = -0.65$] and nature vs. indoor [$b = -0.66$, SE = 0.15, $t_{(48)} = -4.35$, $p < 0.001$ one-tailed, 95% CI = [−0.956, −0.355], $d_{rm} = -0.48$], but again not when comparing urban vs. indoor [$b = 0.17$, SE = 0.15, $t_{(48)} = 1.12$, $p = 0.27$, 95% CI = [−0.135, 0.475], $d_{rm} = 0.13$]. In addition to the preregistered immediate intensity and unpleasantness ratings, participants were asked to assess retrospectively (directly after concluding a complete pain block, i.e., exposure to an environment coupled with painful shocks) to what extent viewing the respective environments helped distract them from or better tolerate the shocks. Exploratory analyses of these ratings revealed a significantly higher level of distraction from and tolerance of the shocks for the nature condition compared to both the urban and indoor conditions, while contrasting the latter two did not yield any differences (Supplementary Results).

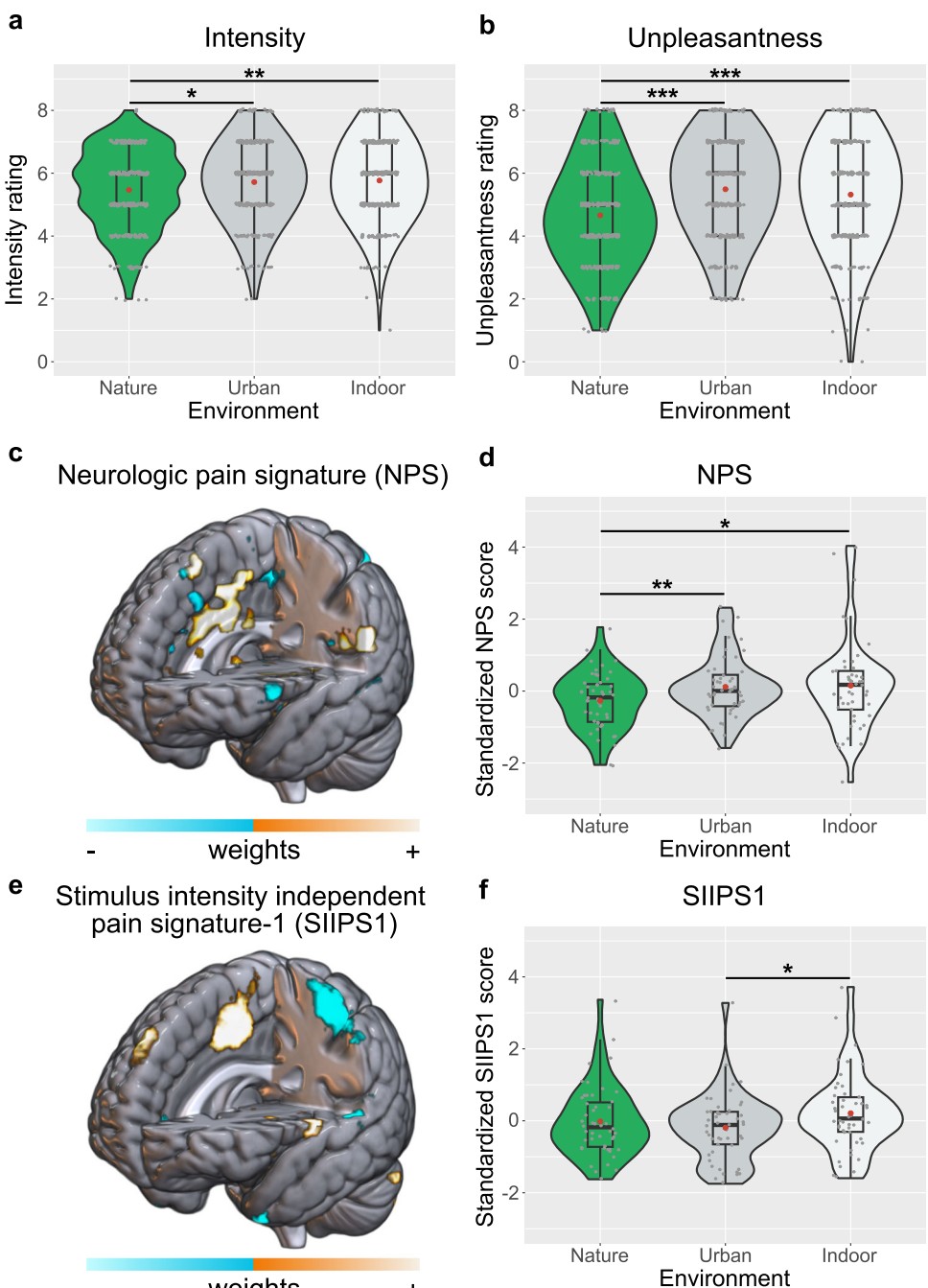

**Fig. 2 | Behavioral and pain signature responses across environments.** Violin plots depicting **a** intensity and **b** unpleasantness ratings ($n = 49$, 6 ratings for each participant and domain per environment; i.e., 1764 overall ratings) of painful shocks and the overall lower-level nociceptive **d** and higher-level cognitive-emotional **f** neural response to pain ($n = 49$, per environment) as indicated by the neurologic pain signature (NPS, **c**) and the stimulus intensity independent pain signature-1 (SIIPS1, **e**). Both brain maps show the signatures' weights (positive = orange, negative = blue). For display purposes, the SIIPS1 map shows weights that exceed a predefined threshold (false discovery rate of $q < 0.05$). Intensity and unpleasantness ratings were given on a scale from 0 ("not at all painful/unpleasant") to 8 ("very painful/unpleasant"). NPS and SIIPS1 responses are plotted as standardized signature scores (see Methods). Gray and red dots in violin plots represent single values (i.e., single ratings for intensity and unpleasantness, and single NPS or SIIPS1 responses) and mean scores, respectively. The boxplots within the violin plots display the median, the first and third quartiles (bounds), and whiskers extending to 1.5x the interquartile ranges from the bounds. Comparing nature to urban environments revealed significant planned pairwise contrasts for intensity ratings ($p = 0.018$), unpleasantness ratings ($p = 0.000002$), and the NPS response ($p = 0.006$; all one-tailed). Similarly, comparisons between nature and indoor environments yielded significant planned pairwise contrasts for intensity ratings ($p = 0.005$), unpleasantness ratings ($p = 0.00004$), and the NPS response ($p = 0.010$; all one-tailed). Additionally, comparing urban to indoor environments revealed a significant planned pairwise contrast for the SIIPS1 response ($p = 0.011$, two-sided). Planned pairwise contrasts were based on the mixed effects models (unadjusted p-values; all significant comparisons remain significant after applying Bonferroni-Holm corrections, see Supplementary Information). Source data are provided as a Source Data file.

These results confirm our preregistered hypotheses and go beyond prior findings of self-reported pain reduction. They indicate that the change in pain is specific to a decrease in the natural setting, rather than an increase in pain for the urban setting. Furthermore, within the typical limitations of self-report, the immediate ratings suggest that both sensory-discriminative (indicated by intensity) and affective-motivational (indicated by unpleasantness) processing were impacted similarly, but that the latter showed a more pronounced effect. The exploratory findings of the retrospective ratings provide the additional insight that participants perceived nature stimuli as helping them with pain tolerance via attention distraction.

### Nature stimuli reduce nociception-related neural responses to pain

We first clearly confirmed that the pain paradigm effectively engaged brain signatures and regional responses classically associated with neural pain processing (see Supplementary Results showing significant NPS, SIIPS1, and region of interest responses for pain vs. no pain across all three conditions). We then assessed the main hypothesis that exposure to nature vs. control stimuli differently affects multivoxel signatures of lower-level nociception-related or higher-level cognitive-emotional responses to pain. To this end, we first computed the NPS and the SIIPS1 in each environmental condition and then compared them using LMM with the signatures per condition as the dependent variable. We found no significant result for the main effect of environment [$F_{(2,48.00)} = 2.29$, $p = 0.111$)], but, importantly, a significant interaction effect of environment*signature [$F_{(2,96.00)} = 5.23$, $p = 0.007$], indicating that the environments impacted the NPS and SIIPS1 differently. Specifically, planned pairwise contrasts revealed significant decreases in the NPS response during nature compared to urban [$b = -0.38$, SE = 0.15, $t_{(96.9)} = -2.55$, $p = 0.006$ one-tailed, 95% CI = [−0.683, −0.085], $d_{rm} = -0.47$] and indoor environments [$b = -0.42$, SE = 0.18, $t_{(79.3)} = -2.36$, $p = 0.010$ one-tailed, 95% CI = [−0.776, −0.066], $d_{rm} = -0.38$], with low to moderate effect sizes. There was no significant effect when comparing urban vs. indoor environments [$b = -0.04$, SE = 0.16, $t_{(92.5)} = -0.23$, $p = 0.81$, 95% CI = [−0.346, 0.273], $d_{rm} = -0.03$]. For the SIIPS1, no significant effects for the nature vs. urban or indoor comparison were found ($p = 0.87$ and $p = 0.11$, both one-tailed; Supplementary Results), but a significant difference of urban vs. indoor [$b = -0.40$, SE = 0.16, $t_{(92.5)} = -2.58$, $p = 0.011$, 95% CI = [−0.712, −0.093], $d_{rm} = -0.40$] (see Fig. 2d, f). Importantly, these effects remained largely consistent after excluding statistical outliers, participants exceeding motion thresholds, and when applying an alternative first-level model specification for the MRI data (Supplementary Results).

The signature-based analyses provided important insights into how the three different environments affected comprehensive neural activation patterns related to pain. Inspired by recent multiverse approaches of neuroimaging data[32] aiming to identify converging evidence across complementary analysis approaches, we had planned and preregistered additional analyses of specific regions of interest (ROIs) and how their activation was affected by the three environments. Selection of the ROIs was theory-based, covering key areas of three circuits involved in the processing and modulation of pain (Methods and Supplementary Fig. 2) identified in an influential framework for pain research[21]. The first circuit represents the ascending pathway and includes the primary somatosensory cortex (S1) and the thalamus. The two other circuits represent descending modulatory systems engaged by psychological pain alterations. One circuit encompasses the superior parietal lobe (SPL), secondary somatosensory cortex (S2), posterior insula (pINS), and amygdala and is associated with attentional modulations of pain. The other circuit covers the anterior insula (aINS), anterior midcingulate cortex (aMCC), medial prefrontal cortex (mPFC) and periaqueductal gray (PAG) and is engaged when emotions alter pain.

Analyzing each of the ROIs separately using a LMM revealed the following significant results for the main effect of environment: thalamus [$F_{(2,48.00)} = 6.04$, $p = 0.005$], S2 [$F_{(2,48.00)} = 7.84$, $p = 0.001$], pINS [$F_{(2,47.99)} = 18.61$, $p < 0.001$], SPL [$F_{(2,48.00)} = 3.98$, $p = 0.025$] and a trend for the aINS [$F_{(2,48.00)} = 3.08$, $p = 0.055$]. Planned pairwise contrasts for these ROIs (Supplementary Fig. S3) revealed a significant difference when comparing nature vs. urban in the thalamus [$b = -0.28$, SE = 0.11, $t_{(48)} = -2.55$, $p = 0.014$ one-tailed, 95% CI = [−0.491, −0.058], $d_{rm} = -0.41$], S2 [$b = -0.46$, SE = 0.12, $t_{(48)} = -3.78$, $p < 0.001$ one-tailed, 95% CI = [−0.707, −0.217], $d_{rm} = -0.53$], pINS [$b = -0.91$, SE = 0.15, $t_{(48)} = -6.09$, $p < 0.001$ one-tailed, 95% CI = [−1.206, −0.608], $d_{rm} = -0.93$, $d_{rm} = -0.93$], and a trend in the SPL [$b = -0.54$, $t_{(48)} = -1.87$, $p = 0.067$ one-tailed, 95% CI = [−1.11, −0.056], $d_{rm} = -0.29$]. Comparing nature vs. indoor revealed a significant difference in the thalamus [$b = -0.39$, SE = 0.12, $t_{(48)} = -3.41$, $p < 0.001$ one-tailed, 95% CI = [−0.626, −0.162], $d_{rm} = -0.53$], S2 [$b = -0.49$, SE = 0.17, $t_{(48)} = -2.89$, $p = 0.001$ one-tailed, 95% CI = [−0.827, −0.149], $d_{rm} = -0.46$], pINS [$b = -0.49$, SE = 0.19, $t_{(48)} = -2.49$, $p = 0.016$ one-tailed, 95% CI = [−0.897, −0.096], $d_{rm} = -0.43$], the SPL [$b = -0.97$, SE = 0.36, $t_{(48)} = -2.75$, $p = 0.013$ one-tailed, 95% CI = [−1.690, −0.261], $d_{rm} = -0.44$] and the aINS [$b = -0.43$, SE = 0.17, $t_{(48)} = -2.45$, $p = 0.036$ one-tailed, 95% CI = [−0.789, −0.078], $d_{rm} = -0.37$]. None of the remaining ROIs showed significant differences for the main effect of environment (all $p > 0.1$, Supplementary Results). Calculating planned pairwise contrasts between urban vs. indoor for the ROIs reported above also revealed no significant differences (Supplementary Results). Importantly, the effects in the thalamus, S2, and pINS remained largely consistent after excluding statistical outliers, participants exceeding motion thresholds, when applying an alternative first-level model specification for the MRI data, and when using alternative ROI masks (Supplementary Results). The effects observed in the SPL and aINS were more sensitive to these analytical choices and should, therefore, be interpreted with caution. Complementary exploratory whole-brain analyses examining activity outside the preregistered ROIs revealed significant voxels only in the primary auditory cortex, when contrasting pain>no-pain in the urban compared to the nature condition (Supplementary Results).

In summary, the multivoxel and region of interest analyses converged in showing that pain responses when exposed to nature as compared to urban or indoor stimuli were associated with a decrease in neural processes related to lower-level nociception-related features (NPS, thalamus), as well as in regions of descending modulatory circuitry associated with attentional alterations of pain that also encode sensory-discriminative aspects (S2, pINS).

Note that in addition to these analyses addressing our hypotheses related to pain outcomes, we had preregistered three additional hypotheses. In brief, we observed (1) that environments significantly differed regarding positive and negative affect (with nature showing higher positive and lower negative affect ratings when compared to urban or indoor settings), (2) that pulse rate was lower in the nature than in the urban setting, and (3) that nature connectedness, contrary to our prediction, did not moderate the main findings (i.e., participants who felt more psychologically close to nature did not show greater benefits compared to those who felt less connected). We also performed further exploratory analyses on the association between self-reported and neural pain responses (including the association of difference scores across environment pairs between immediate ratings and neural responses), as well as the role of immersion. Details on these analyses, the results, and their interpretation are documented in the Supplementary Results.

## Discussion

This preregistered neuroimaging study investigated whether exposure to nature vs. urban or indoor control stimuli mitigates subjective and neural responses to acute pain. Using carefully designed and

controlled stimuli and leveraging neuroimaging techniques, we aimed to address two potential major confounds of previous findings. First, that differences in aesthetic appeal and aversive features of the contrasting stimuli rather than the positive qualities of the nature stimuli explained the observed changes in pain. Second, that constraints associated with subjective pain measures, such as reporting biases or experimental demand effects, confounded earlier results. Furthermore, drawing upon a comprehensive preregistered analysis approach of the fMRI data enabled us to specifically identify which neural responses to pain were predominantly affected by nature exposure.

Following this approach, we demonstrate that natural settings, compared to matched urban or indoor scenes, induce genuine analgesic effects, that the effects are likely positive consequences of the nature stimuli rather than being caused by the aversiveness of the standard 'urban' control stimuli, and that this effect can be primarily attributed to changes at sensory and nociception-related lower levels of the processing hierarchy. More specifically, nature exposure was associated with a reduced response in a highly precise and sensitive neurological signature of pain (the NPS) linked to nociception-related brain processes[27]. Complementary univariate analyses showed lowered pain-related activation in areas receiving nociceptive afferents (thalamus, S2, pINS), providing converging evidence that nature exerted its effects on areas predominantly associated with lower-level sensory pain components. Moreover, the stimulus-intensity independent pain signature-1 (SIIPS1), used to capture higher-level pain-related processes, was not differentially affected by the nature stimuli. While this further supports the idea that nociception-related rather than cognitive-emotional aspects underpinned the subjective analgesic effects, it is important to note that the SIIPS1 analysis was not preregistered and should, therefore, be interpreted more cautiously.

Generally, the neural findings confirm the majority of our preregistered hypotheses, which had been conceived to address and replicate past research based on pain self-report[10,12,33]. Regarding our first set of hypotheses, we crucially extend the specificity of previous findings by demonstrating that comparing virtual nature to a matched urban and an additional neutral indoor scene leads to consistent patterns of reduced self-reported pain. Including two control conditions and showing that pain ratings were lower in the nature setting (but similar in the urban and indoor scene), we find that alterations in pain are attributable to a decrease in the nature condition rather than an increase in the urban one—a confound that seems particularly plausible as most urban environments are associated with increased stress levels[8]. Importantly, unlike most past work, we used pre-tested and published stimuli of closely matched natural and urban settings both of which have been rated comparably in terms of perceived beauty[34]. Specifically, the urban stimuli contained many appealing and attractive elements from the nature scene, reducing the possibility that any differences would result from merely creating a spatially unmatched, noxious, and aesthetically unpleasing urban setting[13,14]. Nevertheless, we cannot rule out the possibility that the urban environment may have been perceived as generally aversive. Importantly, however, the self-reported and neural pain responses to the urban setting were broadly comparable to those in the more "neutral" indoor setting. This indicates that the predominant driver of differences in pain processing between the nature and urban settings was not the assumed higher aversiveness of the urban environment. Rather, it suggests that the observed pain differences between the urban and nature settings stem from the positive effects of the nature setting. Further support for this interpretation comes from the consistent patterns across both immediate and retrospective pain ratings, suggesting a similar pattern of differences across conditions.

The consistency of immediate and retrospective ratings is also important because it convergently validates the experimental effects and reveals important intuitions and introspective insights by the participants into how the three environments may have influenced their pain experience and its regulation. Specifically, that participants thought the nature scenes helped to distract them from the pain, and in this way, to tolerate the shocks better is an aspect that converges with attention-related neural processes as a possible mechanism of reduced nociceptive pain that we will discuss further below. However, the immediate ratings of intensity and unpleasantness also reveal why it is important to complement self-report using neural data[19]. Indeed, while both types of ratings were lower in the nature setting, effect sizes were higher for unpleasantness than intensity ratings. This suggests that nature possibly influenced the affective-motivational more than the sensory-discriminative components of pain[21], a conclusion that is not supported by the neural findings. A possible explanation of these discrepancies across behavioral and neural data is that the self-report may reflect participants' intuitions about how the different environments will impact their experiences. Since subjective ratings are the result of an intricate interplay between various mechanisms (including nociception, emotion, or cognition), using such ratings alone would make it difficult to conclude which specific aspect of pain processing was impacted[17].

Leveraging highly sensitive neural indicators of specific pain components helped us overcome such limitations. Using these neural indicators demonstrates that the decreased subjective reports of pain are associated with reduced neural responses in lower-level nociceptive pain, as indicated by a selective effect on the NPS. This is a key finding, as the NPS entails several regions that receive nociceptive afferents and shows high pain specificity[19,27]. There is thus broad consensus that experimental manipulations that result in changes of this signature indicate genuinely pain-related, and rather nociception-related, brain states (for a critical account[18]). Importantly, the effects of the environmental conditions on the NPS, similar to the self-report effects, are specifically related to the nature stimuli and not confounded by increases in pain processing due to inappropriately matched urban and indoor control stimuli.

Beyond demonstrating pain specificity, comparing the NPS with another pain signature, the SIIPS1, revealed that nature acted predominantly on nociception-related rather than domain-general aspects of pain. Of note, while the SIIPS1 has been developed to capture pain-related processes as well, in contrast to the NPS it intends to characterize domain-general cognitive and affective aspects engaged during the experience of pain beyond nociception-related and somatosensory processing[28]. Pain regulation or valuation are two examples of such aspects, which are linked to ventral and dorsal prefrontal cortex activity and thus to higher-level associative brain areas farther removed from the direct somatosensory inputs[35]. Therefore, it is noteworthy that this signature, and how it tracked the acute pain we exposed our participants to, was not significantly influenced by the nature vs. control stimuli. However, it is important to approach our interpretation of the SIIPS1 response with caution. On the one hand, viewing the SIIPS1 as reflecting domain-general aspects is contingent upon the shortcomings of previous evidence validating its specificity to pain. Consequently, as suggested elsewhere[30], we propose that – unlike the NPS – the SIIPS1 may be influenced by cognitive and affective processes that are not specifically or exclusively associated with pain. On the other hand, we only preregistered the investigation regarding the NPS but not the SIIPS1 since we had originally planned to disentangle which pain components are predominantly affected using pooled ROI activity. This decision was adopted later, but before looking at the data, because a direct comparison between signatures upon further reflection seemed more parsimonious and valid (see Supplementary Methods for further rationale). Thus, the selective effects on the NPS require further confirmation, as does the specific interpretation of the NPS versus the SIIPS1 in tracking nociception-related vs. domain-general responses engaged during the experience of pain.

That said, the complementary analyses of individual ROIs strengthen the signature-based findings that nature exposure acts on

lower- rather than higher-level pain processing. Of note, these ROI analyses were planned with two rationales in mind. First, in the spirit of multiverse analyses[32], they aimed to analyze our data in different ways and render our conceptual conclusions more convincing if convergent evidence was revealed. Second, they allowed us to tap into distinct pathways connected to pain and its neural representation. Compared to the more data-driven brain signatures, these neural pathways are based on long-standing theoretical accounts grounded in pain physiology and clinical practice[15,21]. Drawing upon these accounts, we find decreased activation during the nature condition in the ascending pathway (thalamus) receiving direct input from nociceptors and a modulatory circuit involving areas associated with sensory-discriminative processing (e.g., S2, pINS). In contrast, brain regions related to a circuit underlying higher-level emotional modulations of pain (e.g., aMCC, mPFC) showed no difference between environments. While the engagement of any brain area during complex experiences, including pain, likely does not adhere to a singular function or a dichotomous functional organization, this remains an important finding. It enables us to disentangle the underlying mechanisms, relate the findings to influential accounts of the benefits of nature from environmental psychology, and put them into perspective relative to other non-pharmacological interventions.

For instance, in the most extensive single neuroimaging study of placebo effects to date, it was suggested that placebo manipulations do not impact nociception-related (NPS), but instead domain-general cognitive-emotional aspects (SIIPS1) of pain[30]. This is in direct contrast to our findings and suggests that nature-related pain reductions are likely not based on belief processes and expectation effects such as the ones investigated by placebo research. Instead, pain relief through nature exposure seems to be more related to changes in sensory circuitries and attentional processes connected to the engagement of these circuits. Similar results have been found among participants engaged in attention-based mindfulness practices[36,37], where training participants in mindfulness practices over eight weeks was associated with changes in lower-level nociception-related (NPS) but not higher-level cognitive-emotional (SIIPS1) responses to pain. The authors interpreted this reduced NPS response as changes in attentional mechanisms that gate lower-level nociceptive signals.

Regarding nature's potential to alleviate pain, the interpretation that reduced NPS activity is indicative of alterations in attention is particularly intriguing. In the field of environmental psychology, two prominent theories provide indirect frameworks for explaining nature's analgesic effects on pain. On the one hand, SRT posits that nature primarily influences affective responses[8]. In the context of pain research, SRT would imply that nature's impact on pain should be explained by altered affect and reduced activity in higher-level neural processing linked to these affective changes (e.g., SIIPS1, PFC, or aMCC responses). As we did not observe such differences, our data offer limited support for such an interpretation. On the other hand, ART suggests that natural stimuli can restore depleted attentional capacities[9]. The reasoning behind this argument is that nature possesses many features that are softly fascinating to humans and engage us in a distracting but not overly demanding manner. In the context of the experience of pain, ART would imply that features of the nature stimuli divert attention away from the painful sensation. In conjunction with findings from neuroscientific pain research, the observed reduction in nociception-related responses (e.g., the NPS, thalamus, S2, and pINS) substantiates this interpretation in favor of ART in two ways.

First, neuroscientific accounts of pain propose that different modulatory neural systems are engaged when pain is altered by emotional or attentional processes[21]. For instance, previous studies have shown that if attention is diverted from a painful stimulus, this is visible in changed responses in areas related to sensory-discriminative processing[38–40] (for a critical review[41]). According to these frameworks, attentional modulations of pain are characterized by pathways involving projections from the superior parietal lobe to the insula, S2, and amygdala[21]. We observed robust effects for two of these areas (pIns, S2), and less robust findings for one (SPL) when comparing nature to urban or indoor stimuli. Second, asking participants if exposure to the respective environment helped to distract themselves from pain revealed effect sizes in the medium to high range when comparing nature to urban ($d_{rm} = 0.66$) or indoor settings ($d_{rm} = 1.04$) while comparing urban and indoor stimuli ($d_{rm} = 0.34$) showed only a small effect (Supplementary Results). Together, these theoretical accounts and our findings render it plausible that the effects on nociceptive signaling and its cortical representations are linked to attention-related processes. Notwithstanding these arguments, it is important to acknowledge that ART, SRT, and pain theories originate from distinct disciplines, each with unique terminologies and research foci. As a result, establishing direct connections between them remains somewhat imprecise at present and will require theoretical refinement and alignment in future work. We note that the attention regulation mechanism and the precise pattern of results were not specifically preregistered. The postulated interaction between attention- and nociception-related processes thus needs confirmation and extension by future research, which should focus on identifying how exactly attention-related brain areas act as regulators of the nociceptive inputs.

Besides these propositions for future work, our findings open several other exciting research avenues. First, participants in our study were not exposed to real-world environments but to virtual stimuli. While this approach allowed us to maximize experimental control, whether the results are generalizable to real-world contexts remains to be tested. That our findings are based on virtual stimuli is a major strength, though. It suggests that nature-based therapies do not necessarily require real-world exposure, but that stimuli acting as proxies for such environments might suffice. This is a particularly promising aspect as it suggests a broad range of use cases that can be employed cost-efficiently in a wide range of interventions.

Second, more granularity is required to thoroughly assess which specific elements of nature are relevant in driving the observed analgesic effects. The literature on the benefits of nature suggests that certain perceptual features make natural settings particularly fascinating[9,13]. These features might exhibit a notably engaging effect, thus leading to a stronger diversion from pain. Complex cognitive and emotional reactions, such as feelings of awe and nostalgia, towards these features might be essential[42], but which particular feature is relevant remains unclear. However, our exploratory whole-brain analysis indicated that, despite being matched in loudness, the urban environment resulted in distinct processing in auditory cortex, compared to nature. This finding provides preliminary evidence that it may not only be the visual quality of the nature stimuli that makes them effective, but that variations in soundscapes may have downstream effects on pain processing as well. Notably, although differences in auditory processing may relate to alterations in pain, they are unlikely to be the primary driver of the observed effects. This is evidenced by the lack of differences in auditory processing between the nature and indoor environments, despite comparable magnitudes of change in pain outcomes across both the urban vs. nature and indoor vs. nature comparison. Thus, further work is needed to explore which specific sensory elements and their combination make natural environments particularly effective in alleviating pain.

Third, while we highlight the limitations of subjective pain measures in previous studies due to potential reporting biases, our own self-reported data faced similar shortcomings. However, the convergence between neuroimaging and self-report findings instills added confidence that the present and prior findings are not predominantly driven by such biases. Additionally, since the neural results contrast with those from placebo studies[30,36], it seems less likely that they are entirely attributable to expectation or demand characteristics.

Fourth, while harnessing neuroimaging enabled us to interrogate the effects of natural settings on pain processing with high specificity, some accounts challenge the notion that neuroimaging indicators can entirely dissociate pain from other phenomena[18]. Furthermore, dichotomizing brain regions as either sensory-discriminative ('lower-level') or affective-motivational ('higher-level') may oversimplify their roles, which are often multifaceted. Indeed, some authors challenge this separation, highlighting that evidence for distinct modulations of these components remains inconclusive[22]. Thus, while caution is warranted when interpreting regional and patterned brain activations, the convergence across neuroimaging analyses strengthens the argument that the nature stimuli primarily affected 'lower-level' components.

Finally, considering the severe impact chronic pain has on patients and our society and the potential risks associated with its pharmacological treatment, nature exposure represents an interesting complementary pain management strategy. While the current study provides first evidence as to which underlying processes are altered in the processing of acute pain, chronic pain is characterized by complex and multifaceted changes in psychological and neural processing[43] that only partially converge with those during acute pain. It is an exciting research avenue to test the generalization of the present findings to chronic pain and the potential alleviation of chronic pain conditions.

In conclusion, our results show that simple and brief exposure to nature reduces self-reported and specific neural responses to acute pain and is linked to lower-level pain-specific nociception-related processing. In contrast to other non-pharmacological interventions, which usually involve complex deceptions through placebo induction procedures or week-long training of cognitive coping strategies, the nature stimuli used here potentially provide an easily accessible alternative or at least complementary intervention in clinical practice. Incorporating natural elements into healthcare design has the potential to reduce pain-associated complaints and constraints with relatively low effort. This is important and promising from a clinical-applied perspective: it suggests that employing natural stimuli could be a cost-effective and easily implementable intervention in pain treatment and related contexts to promote health and well-being.

## Methods

### Participants

The study was preregistered on 12 May 2022 (https://osf.io/t8dqu), conducted according to the seventh revision of the Declaration of Helsinki (2013) and approved by the Ethics Committee of the University of Vienna (EK-Nr. 00729). A total of 53 healthy right-handed human participants fulfilling standard inclusion criteria for neuroimaging studies of pain participated. Based on an a-priori power analysis, a sample size of 48 participants was preregistered (Supplementary Methods). Four participants had to be excluded due to technical problems with the pain stimulator and the scanner, leading to a final sample including 24 female and 25 male participants (Age $\pm$ SD = 25.24 $\pm$ 2.79, range = 20–35). All participants received a reimbursement of €30.

### Experimental procedures

Upon arrival, participants were instructed about the study procedure, gave written informed consent, and completed a pain calibration task. Afterward, they entered the MRI scanner and were alternately exposed to blocks of virtual stimuli each depicting a different environment (5 min), directly followed by blocks showing the same environment accompanied by electrical shocks (9 min; from here on referred to as video and pain blocks, respectively). This design enabled participants to familiarize themselves with each respective environment before its presentation alongside the pain stimuli. The order of the presented environments was counterbalanced across participants. Participants completed several ratings after each block, leading to an

approximately 5 min pause between them. The total duration of the study procedure was 120 min.

To deliver an engaging and immersive experience each environment was created by a dedicated professional graphic designer and depicted a virtual environment accompanied by a matching sounds-cape. Three different environments were presented in counter-balanced order, showing a natural, an urban, or an indoor setting (Fig. 1a). The natural and urban environments were adapted from previously published studies[34,42]. They were closely matched in terms of various visual characteristics, structural proportions, and physical features such as lighting conditions and loudness of sounds. Both environments were designed to be generally favorable by including and matching elements that prior research has identified as appealing, such as a large water body, reflective surfaces, green foliage, and a high level of complexity and openness of space[44–46]. Specifically, the natural setting was created first and included a large central lake (with observable wind ripples), trees by the side of the lake (with rustling leaves), and an animation showing the shifting position of the sun and cloud movements. The urban condition was constructed by adding human-made elements to this basic scene, including buildings on the far side of the lake, a paved path, a short wall, and benches on the nearside of the lake. While the scenes were not deliberately matched based on subjective measures, the resulting urban scene, containing many of the originally attractive natural elements, was still rated as relatively beautiful[34]. Both scenes were accompanied by soundscapes created based on recommendations of previous works investigating acoustic experiences in different environments[47]. The nature scene included the sounds of rippling water, gentle wind, native birds, and insects, while the urban scene included the sounds of different vehicles and construction works. For both environments, careful consideration was given to selecting and adjusting all sounds based on factors such as the nativeness of species, typical local traffic noises (e.g., emergency vehicle horns), or the time of day. The indoor setting depicted a desk with office supplies, a fan, and a computer. It was accompanied by the sounds of a computer and a fan. The soundscapes of all environments were normalized regarding their average loudness by matching the root-mean-square amplitude. During the video blocks, participants observed each scene while being instructed to imagine themselves being present in the specific environment. This was facilitated by reading through a short script immediately preceding each video block. The scripts were based on previous nature-based guided imagery interventions[48].

During pain blocks, participants were instructed to read a short immersion script, then re-watched the same environment from the preceding video block while receiving electrical shocks, with the video playing in the background. Thirty-two electrical shocks (16 painful and 16 non-painful) were administered per block. To ensure comparable pain intensities across participants, the stimuli were calibrated according to an established procedure[49,50]. Painful shocks were calibrated to represent a "very painful, but bearable" (6), and non-painful shocks to represent a "perceptible, but non-painful" (1) sensation on a scale from 0 ("not perceptible") to 8 ("unbearable pain"). The calibration consisted of three phases, separated by breaks of approximately 3 min, and was conducted inside the scanner. Participants received an initial low-intensity shock (0.05 mA) in the first two phases, followed by progressively stronger shocks rated on a scale from 0 to 8. Each phase concluded when a shock was rated as 8, resulting in a variable number of shocks for each participant. In the third phase, shocks reflecting the average intensities of 1 and 6 from the first two phases were pseudorandomly administered and rated by the participants to confirm the stability of their perceived intensity. We administered the shocks using a Digitimer DS5 Isolated Bipolar Constant Current Stimulator (Digitimer Ltd, Clinical & Biomedical Research Instruments). Two electrodes, one for painful and one for non-painful shocks, were

attached to the dorsum of the left hand. Mean shock intensities were 0.61 mA (SD = 0.42) and 0.19 mA (SD = 0.09) for painful and non-painful trials, respectively, which is comparable to previous studies in our laboratory following a similar protocol[49,51]. Each pain block presented the painful and non-painful trials in the same pseudo-randomized order. Pseudorandomization was employed to ensure that the co-occurrence of painful shocks and specific auditory and visual elements of the environments were kept constant across participants and conditions. In line with previous uses of the pain paradigm[49,51], every trial started with a colored visual cue displayed for 2000 ms that indicated the next shock's intensity (painful = red, non-painful = yellow). After a variable pause where the cue disappeared (jittered with 3500 ± 1500 ms), another visual cue was presented for 1000 ms with the electrical stimulus being administered for 500 ms simultaneously. The second visual cue matched the first cue in shape and size but had a colored filling. Next, the cue and shock disappeared for a variable duration (jittered with 3500 ± 1500 ms). An additional intertrial interval of 2000 ms separated all trials (Fig. 1b). Twelve of the 32 trials (six painful and six non-painful) were succeeded by two ratings to indicate the perceived intensity ("How painful was the shock for you?") or unpleasantness ("How unpleasant was the shock for you?") of the last administered shock on a scale ranging from zero ("not at all") to eight ("very"). Notably, the visual cues for each trial were superimposed on the virtual scene, which continuously played in the background to maximize the immersion into the environment. The visual and accompanying audio stimuli were presented on an MRI-compatible 32-inch display (Full HD 1920 × 1080 PPI resolution; BOLDscreen 32 LCD, Cambridge Research System, Cambridge, UK) viewed at 26° × 15° visual angle, and Sensimetrics earphones (model S14; Sensimetrics Corporation, Gloucester, MA, USA), respectively. All stimuli and ratings were presented using MATLAB R2021a (Mathworks, 2021) and Psychophysics Toolbox Version 3[52].

## fMRI acquisition, preprocessing, and analysis

fMRI data were acquired with a 3 Tesla Siemens Magnetom Skyra MRI scanner (Siemens Medical, Erlangen, Germany). The scanner was equipped with a 32-channel head coil. Each run acquired a separate functional volume using a multiband-accelerated gradient echo echoplanar imaging sequence, for one of the three pain blocks using the following parameters: Repetition time (TR) = 800 ms, echo time (TE) = 34 ms, flip angle = 50°, field of view (FOV) = 210 × 210 × 138 mm³, multi-band acceleration factor = 4, interleaved multi-slice mode, interleaved acquisition, matrix size = 96 × 96 × 36, voxel size = 2.18 × 2.18 × 3.84 mm³, 36 axial slices of the whole brain with slice thickness = 3.50 mm and an interslice gap of 0.34 mm. We used a magnetization-prepared rapid acquisition gradient echo sequence with the following parameters to obtain the structural image at the end of each scanning session: TR = 2300 ms, TE = 2.29 ms, flip angle = 8°, FOV = 165 × 240 × 240 mm³, ascending acquisition, single shot multi-slice mode, 176 sagittal slices, matrix size = 176 × 256 × 256, voxel size = 0.94 × 0.935 × 0.935 mm³, slice thickness = 0.94 mm. Furthermore, field map images were acquired using a dual-echo gradient echo sequence to correct the functional images for magnetic field inhomogeneities, with the following parameters: TR = 400 ms, TE1 = 4.92 ms, TE2 = 7.38 ms, flip angle = 60°, FOV = 220 × 220 × 138 mm³, matrix size = 128 × 128 × 36, voxel size = 1.72 × 1.72 × 3.84 mm³, 36 axial slices aligned with the orientation of the functional images, and slice thickness = 3.84 mm. Field map correction was checked visually by comparing corrected and uncorrected images and inspecting potential distortions in areas that are prone to artifacts (e.g., near the sinuses, edges of the brain).

Preprocessing of the fMRI data was performed using SPM12 (Wellcome Trust Centre for Neuroimaging, www.fil.ion.ucl.ac.uk/spm) running on MATLAB 2021a (Mathworks, 2021), including the following steps: realignment and unwarping using participant-specific field maps, slice-time correction with the center slice as

reference, coregistration of functional and structural images, segmentation into three tissue types (gray matter, white matter, cerebrospinal fluid), spatial normalization to Montreal Neurological Institute space using Diffeomorphic Anatomical Registration Through Exponentiated Lie Algebra (DARTEL), and spatial smoothing with a 6-mm full-width at half maximum 3D Gaussian Kernel. The first-level analyses followed a general linear model (GLM) approach. A design matrix was specified with the following five experimental regressors per environment (i.e., run): anticipation of painful shocks, anticipation of non-painful shocks, delivery of painful shocks, delivery of non-painful shocks, and rating. Furthermore, six nuisance regressors from the realignment step accounting for movement-induced noise were added. The experimental regressors were time-locked to the onset of each trial phase and convolved using SPM12's standard hemodynamic response function in an event-related fashion. Furthermore, we applied SPM12's standard temporal filter methods, including the use of a high-pass filter with a default cut-off of 128 s to remove low-frequency noise.

To ascertain that our pain paradigm, as expected and extensively demonstrated in prior work[20,27,28,53], robustly activated single-region and multivariate signature responses to pain, we first performed an analysis that was orthogonal to our main hypotheses. This analysis revealed conclusive evidence that our pain task evoked neural activity in pain-related brain regions (e.g., bilateral thalamus, bilateral S2, insula, and amygdala), the NPS and SIIPS1, and all preregistered ROIs (except for the S1, which was no longer significant after applying conservative corrections for multiple comparison, Supplementary Results). Therefore, we proceeded to test our main hypotheses on whether these neural responses to pain are reduced by exposure to nature. To this end, one contrast image was created comparing pain > no-pain trials for each environment. First, we investigated whether the overall lower-level nociception-related and higher-level cognitive-emotional neural response to pain differed for each environment by applying the NPS and the SIIPS1 to our first-level GLM beta maps[27]. This was done using scripts created by the developers of these patterns[27,28], which were made available to us after personal inquiry. We calculated the dot product of the contrast image and the pattern map of the NPS and SIIPS1, resulting in two scalar values for each participant and environment. The NPS and SIIPS1 represent multivoxel patterns within and across pain-related brain regions that track lower-level or higher-level pain processing, respectively[27,28]. Second, we performed ROI analyses to test our hypotheses using a different methodological approach and to further differentiate if the alterations in pain are predominantly found in areas associated with lower-level or higher-level pain processing. We created the following preregistered set of sphere-based ROIs (center [± x, y, z]; sphere size; Supplementary Fig. S2): amygdala ([± 20, −12, −10]; 10 mm), anterior midcingulate cortex (aMCC; [−2, 23, 40], 10 mm), anterior insula (aINS; [± 33, 18, 6]; 10 mm), posterior insula (pINS; [± 44, −15, 4]; 10 mm), medial prefrontal cortex (mPFC; [7, 44, 19]; 10 mm), primary somatosensory cortex (S1; [± 39, −30, 51]; 10 mm), secondary somatosensory cortex (S2; [± 39, −15, 18]; 10 mm), periaqueductal gray (PAG; [0, −32, −10]; 6 mm), superior parietal lobe (SPL; [± 18, −50, 70]; 10 mm), and thalamus ([± 12, −18, 3]; 6 mm). Each ROI's center coordinate and sphere size were based on previous meta-analytic findings and pain studies from our lab experimentally inducing acute pain using similar methods[49,51,54]. For each ROI, we only included voxels that showed a significant response to painful vs. non-painful stimuli in the pain>no-pain contrast across environments (see Supplementary Results for ROIs encompassing voxels sensitive and insensitive to pain). Then, we extracted the mean percent signal change per participant for the pain>no-pain first-level contrasts for each individual environment using the MarsBar toolbox[55].

## Statistical analysis

To test our main hypothesis, which was that exposure to nature stimuli reduces self-report and neural responses to pain, we ran several LMMs

using the lmer function of the lme4 package in R (R Core Team, 2023)[56]. We preregistered the majority of the models (https://osf.io/t8dqu) and specified each of them using maximal random effects structures[57]. For the immediate self-reports on pain, we specified the intensity and unpleasantness ratings of the painful shocks as the dependent variable to be predicted by the fixed effect of environment (nature as the reference), rating content (intensity as the reference), and their interaction (with random slopes and intercepts for environment, rating content and their interaction by participant). For the neural signatures, we used the standardized signature response of the NPS and SIIPS1 as the dependent variable to be predicted by the fixed effect of environment (nature as a reference), signature (NPS as reference), and their interaction (with random slopes and intercepts for environment and signature by participant). Standardization across responses was performed separately for each signature, pooled across environments. For ROIs in one hemisphere, we used the ROI response as the dependent variable to be predicted by the fixed effect of environment (nature as a reference, with random intercepts for participants). For ROIs with spheres in both hemispheres, we used the ROI responses of both hemispheres as the dependent variable to be predicted by the fixed effect of environment (nature as a reference), hemisphere (left as reference), and their interaction (with random slopes and intercepts for environment and hemisphere by participant). For each LMM, we report significance testing for the main effects of environment and interaction effects of interest, followed by planned pairwise contrasts. The p-values of the pairwise contrasts from the ROI analysis were Bonferroni-Holm corrected (separated by the different descending modulatory (attention vs. emotion) and ascending pain circuits; all reported p-values represent adjusted values). For each pairwise contrast, we computed the repeated standardized mean difference (drm) as an effect size using the means and standard deviations of each environment[58]. We interpreted effect sizes based on widely used conventions[59], where small effects are defined as 0.2, medium effects as 0.5, and large effects as 0.8. An exemplary model syntax, using the response in the S2 as a dependent variable, looked like this:

$$S2_{response} \sim 1 + environment * hemisphere$$
$$+ (1 + environment + hemisphere | participant)$$

Details regarding all models (e.g., formulae, model fit, random effects variance and correlation, etc.) and deviations from the pre-registration are reported in the Supplementary Information.

### Reporting summary
Further information on research design is available in the Nature Portfolio Reporting Summary linked to this article.

## Data availability
The behavioral data, region of interest and multivariate signature data extracted from the fMRI signal time course, thresholded whole-brain maps comparing pain>no-pain in urban vs. nature and indoor vs. nature environments, as well as unthresholded statistical maps for the pain>no-pain contrast in each environment, have been deposited on OSF and are accessible at https://osf.io/t8dqu/. Additionally, the Source Data for all Figures and Tables are provided as separate Source Data files. Source data are provided with this paper.

## Code availability
The code for the results of the behavioral, region of interest, and multivariate signature data analyses is accessible at https://osf.io/t8dqu/. The code is divided into two scripts: one corresponding to the results presented in the main text, and another for the analyses included in the Supplementary Information.

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

## Acknowledgements

This research was funded by the Austrian Science Fund (FWF) "DK Cognition and Communication 2": W1262-B29 [10.55776/W1262], and in part by the "Lise Meitner Fellowship": M3166 [10.55776/M3166], and the "Neuronal circuits in health and disease" grant: COE16 [10.55776/COE16]. MW's time on this project was supported by the EU's Horizon Europe research and innovation programme under grant agreement No. 101081420 (RESONATE). LZ's time on this project was partially supported by the Wellcome Trust (228268/Z/23/Z). Participants were, in part, recruited through the Vienna CogSciHub: Study Participant Platform (SPP), based on the Hamburg Registration and Organization Online Tool (hroot; Bock et al., 2014). We thank Magdalena Boch and Ronald Sladky for their support with MRI-related questions, and Sarah Koppel for her assistance with data collection. Additionally, we thank Tor Wager and the Canlab team for providing the code used to analyze multivariate pattern responses to pain.

## Author contributions

M.O.S. conceptualization, methodology, software, validation, formal analysis, investigation, data curation, writing – original draft, writing – review & editing, visualization, project administration. M.P.W. conceptualization, methodology, writing – original draft, writing – review & editing, supervision. L.L. formal analysis, resources, writing – review & editing. L.Z. methodology, formal analysis, resources, writing – review & editing. A.J.S. resources, writing – review & editing. S.K. writing – review & editing. C.L. conceptualization, methodology, resources, writing - original draft, writing - review & editing, supervision, funding acquisition.

## Competing interests

The authors declare no competing interest.
