## [Transparent Peer Review file · Nature Communications]

Nature exposure induces analgesic effects by acting on nociception-related neural processing

Corresponding Author: Professor Claus Lamm

Version 0:

Reviewer comments:

Reviewer #1

(Remarks to the Author)

Response to nature exposure induces hypoalgesia by acting on nociception-related neural processing

In this manuscript, Steininger and colleagues examined the impact of nature exposure on the perception and neural processing of pain in a sample of 49 healthy participants. Their findings demonstrate that immersion in a virtual natural environment significantly reduces subjective measures of pain perception, including intensity and unpleasantness. Moreover, the study observed a decreased expression of the Neurologic Pain Signature (NPS)—a well-established multivariate brain signature indicative of pain driven by nociception—under the nature condition compared to control conditions. However, no significant changes were found in the SIIPS1, a signature associated with cognitive-evaluative mechanisms influencing pain perception.

This study is both novel and original, addressing a highly relevant topic with significant translational implications. Given that current pharmacological treatments for (chronic) pain conditions are often inadequate or even harmful—particularly in the case of opioids—this research offers valuable insights into alternative therapeutic approaches. The study adheres to the highest standards of open and reproducible science, including preregistration, sharing of processed data, and code sharing, which is highly commendable.

I enjoyed reading the introduction and discussion section, which are written very elaborately and comprehensive. However, several parts of the methods and results section could benefit from adding further information. I recommend that the authors address the following points to allow for a more thorough assessment of the validity of the results:

Major points:

The authors complement the NPS analysis with pre-registered region-of-interest (ROI) approaches in predefined brain regions (e.g., thalamus, posterior insula). These analyses appear to corroborate the NPS findings, supporting the notion that the analgesic effects of nature interfere with nociceptive and sensory processing rather than higher-level pain processing. However, the analyses in these predefined brain regions are limited to voxels sensitive to the pain vs. non-painful contrast, which introduces circularity with respect to changes in pain due to increased stimulus intensity.

For greater transparency, the authors should: a) avoid restricting the ROI analyses to pain vs. non-pain-sensitive voxels, and b) provide the univariate whole-brain maps for the relevant contrasts (nature vs. control conditions) to ensure that significant activity differences are not present outside the ROIs.

Additionally, to further substantiate the claim that the analgesic effects of nature are indeed linked to changes in nociception, the authors should include correlational analyses between individual behavioral analgesic effects and neural responses – both for the ROIs and the NPS.

The authors report 5 out of 8 of the preregistered hypotheses. This should be made transparent in the manuscript and explained, why relevant information on arousal, immersion etc are not reported. The authors may want to reconsider including the results on the autonomic data which could strengthen the authors interpretation that nature induced analgesic effects are associated with arousal and attention.

Other points:

Introduction:

In the introduction mentions reporting bias as a potential confounder in prior research. Could the authors elaborate on how this issue was addressed in their design (e.g., instructions given to participants)? If this was not addressed, it should be acknowledged as a limitation.

Methods:

Matching Procedure: The authors emphasize that previous studies on the effects of nature exposure often used suboptimal designs and highlight the importance of incorporating a well-matched control condition. They also claim that the videos used in their study were "carefully matched." Could the authors provide a more detailed explanation of the matching procedure, including the criteria for matching (e.g., subjective measures, physical properties)? Additionally, please specify whether the images were sourced from previously published studies.

The author mention in the main text a video and a pain block, however, no information about the video block (condition order, timing). The preregistration details it, but it would be beneficial to the reader to mention in the main text as well. Were the scripts, to enhance immersion, read before each condition? How much pause were between runs? How long was the total measurement?

The authors refer to their previous work regarding the pain calibration procedure. At least brief information on this and where it was performed (in/outside scanner) should be added to the supplement.

fMRI Modelling: Please specify which regressors were included in the first-level general linear model (GLM), such as anticipation, stimulation, rating, etc.

In the preregistration, outliers are defined as values higher than 3 standard deviations from the mean. Based on Fig2D,F, it seems that the data of the NPS and SIIPS1 still contain outliers. Please specify why the authors deviated from the preregistered criteria and provide complementary analyses with these exclusions.

Were there any participants who were excluded based on motion criteria? What was the exact motion criteria?

Please specify the used fMRI sequence (e.g.: 2D GRE EPI) Please double check the provided parameters (there is a mismatch between the fMRI's FOV, matrix size, and voxel size). Specify the sequence for field map acquisition. How was the field map correction checked?

Results

One of the behavioural ratings (intensity) has a value of 4.2. (Fig2A, indoor condition). As the rating was measured in an ordinal scale, this is not plausible. Please systematically check all data and correct.

LI 170 and follows: Modelling of the behavior data resulted in a significant interaction effect. The current phrasing of the meaning of this interaction is a bit confusing ("interaction reflected that the magnitude, but not the overall pattern"). Please clarify.

Effect size calculation: In the calculation of the reported effect sizes, the average within subject variance in each condition was used instead of the within condition variance. This may lead to an overestimation of the reported effect size of the behavioural outcomes (intensity and unpleasantness).

The inference made from these two estimates would not change, but for completeness I would recommend to correct it. Additionally, the calculated effect sizes estimated from a simple repeated measure (basically a paired t test). There is no consensus how to report standardized effect sizes from mixed model, therefore, the authors approach can be accepted. However, it would raise the necessity of the use of a (complex) mixed model, as the main conclusion are based on preregistered directed tests.

Did the authors control for the level of immersion in the different conditions? If participants more immersed in the nature, as compared to the other conditions this would be an important bias, or at least mechanistic explanation.

Figure 1: the timing under the Video should be 3.5+/- 0.5.

Discussion:

The authors may want to consider discussing the role of prior experiences and learning mechanisms in the effect of nature on well-being, as well as the role of learned differences in social aspects in the different conditions.

The authors should temper their conclusions regarding the non-significant effects on the SIIPS1, as this outcome was not part of the original preregistration and it is debatable to draw conclusions from the lack of significance.

In the discussion, the authors suggest that the positive effects of nature stimuli are due to the nature of the stimuli themselves, rather than the aversiveness of the urban control stimuli. However, I did not find data in the current report to support this claim. Was the aversiveness of the urban images measured? If so, could the authors provide additional data or clarification to support this assertion?

(Remarks on code availability)

Reviewer #2

(Remarks to the Author)

The manuscript presents a preregistered fMRI study (N = 49) investigating the behavioral and neural understanding of the analgesic effects of nature exposure on pain induced by electrical stimulation. The pre-registration of this study is commendable, as it strengthens its transparency and credibility. The authors employed well-controlled audiovisual stimuli to compare the effects of nature exposure with those of urban and indoor (neutral) virtual environments. They evaluated the impact of nature exposure on both neural and behavioral responses to noxious stimuli, finding that nature-related audiovisual stimuli (i.e., nature video and sound) reduced pain intensity and unpleasantness ratings compared to other control conditions, urban and neutral stimuli. Furthermore, analyses using fMRI pattern-based markers and ROIs revealed that these analgesic effects were associated with the changes in the main nociceptive processing system in the brain. Overall, this study provides evidence for the brain modulation effects of natural exposure for pain modulation. There are a few things that can improve the study, though.

Major comments:

- The connection between the paper's theoretical focus and the neuroimaging results could be improved. While the authors frame their research around the ART and SRT, suggesting that attentional or affective processes are key to understanding nature's analgesic effects, it remains unclear how the fMRI markers (NPS and SIIPS1) map onto these cognitive processes.
- Relatedly, the authors interpret the NPS as nociception-related and SIIPS1 as domain-general cognitive-emotional. However, how do they ensure that this interpretation is correct? The original paper by Woo et al. (2017) does not clearly make this distinction. Rather, SIIPS1 is described as predicting pain above and beyond nociceptive input and mediating the pain-modulating effects of psychological factors such as expectations and perceived control. The interpretation of SIIPS1 as "domain-general cognitive-emotional" seems to require further justification or evidence beyond what is provided by the original paper.
- In addition, the authors also dichotomize the brain regions' functions: pINS and S2 as early low-level nociception-related processing vs. aMCC and PFC as high-level emotional and motivational aspects. This dichotomy oversimplifies the complexity of these regions' roles in pain processing. For example, pINS and S2 are also included in SIIPS1. Similarly, areas like the PFC and aMCC are not exclusively linked to emotional and motivational processing but also play roles in sensory discrimination and attentional modulation. A more nuanced interpretation would reflect the integration of both sensory and affective processes within these regions, rather than a strict dichotomy.
- Please add figures for ROI analyses. At least the authors should show the location of ROIs to interpret their results more thoroughly.
- In the context of Stress Recovery Theory (SRT), positive affect is a key component, and thus the preregistered plan includes analyses on positive affect (e.g., using PANAS), but nothing related to positive affect is presented in the current manuscript. Given that positive affect may play an important role in the observed effects, particularly in supporting the SRT framework, it would be beneficial to include these results to provide a more comprehensive analysis.
- Relatedly, although the main findings focus on the differential effects between two fMRI pattern-based markers, only one marker (NPS) was preregistered. Furthermore, the relevant hypotheses in the preregistered plan were marked as 'non-directional,' indicating a significant portion of the study is still exploratory. This reduces the strength of the preregistration's contribution. For greater transparency, it would be helpful for the authors to clearly specify which analyses adhered to the preregistered plan, which were exploratory, and which preregistered analyses were omitted in the current study (e.g., positive affect and physiological responses, etc.).
- The connection between the main findings of the current study (e.g., NPS and SIIPS results) and some behavioral findings, such as distraction and tolerance, is not clearly established or explained.
- Furthermore, to elucidate the detailed neural mechanisms of the analgesic effects induced by nature exposure, it would be beneficial to analyze other events (e.g., cue or video) within the experiment. This could help clarify whether the observed effects are stimulus-specific or persist throughout the trials.

Minor comments/questions:

- I am not sure if "hypoalgesia" is the correct term for the pain-modulatory effects of nature exposure. Hypoalgesia is often used in clinical or pharmacological contexts. An alternative term could be "analgesic effects," which is more generally used to describe the reduction in pain intensity or unpleasantness without necessarily implying a clinical or pharmacological mechanism.
- While Fig. 1B effectively outlines the experimental paradigms, it appears that the graphical illustration does not fully capture specific elements of the figure legend, e.g., "after each third trial, participants rated the shock's intensity and unpleasantness at 6,000 ms each". Please consider revising the figure to better align with the described procedures.
- In the results section, the interaction between environment and rating type for pain ratings was reported with degrees of freedom that include decimal points [$F(2,81.14) = 9.19, p < 0.001$], while similar analysis (interaction between environment and signature) with signature responses did not [$F(2,96) = 6.04, p = 0.003$]. If these discrepancies result from different statistical assumptions (e.g., equal variance) applied across the linear mixed model (LMM) analyses, it would be helpful to clarify this in the methods or results sections.
- The manuscript currently lacks detailed information on the pain calibration task (e.g., the number of stimuli and the range). Please provide a description of the pain calibration task in the Method sections.
- Regarding the first-level analysis, could you clarify whether only shock events were included in the design matrix, or if other event-related regressors (e.g., cue presentations and ratings) were also included in the design matrix? Additionally, please specify whether temporal filtering was applied or not.
- Given the potential differences in visual and acoustic features between indoor, nature, and urban stimuli (especially between indoor and other stimuli), were any nuisance regressors included to account for these variations? Clarification on this point would help with the interpretation of the results.
- According to the legend of Figs. 2A and 2B, it is addressed that each dot in the violin plots represents single values. Does

the dot represent immediate self-report ratings for each trial? Please clarify.

- In Figs. 2D and 2F, the standardized betas are displayed rather than the signature response values. Could you provide a rationale for this choice?

(Remarks on code availability)

I have not tried to run it, but the code (R code to run statistical analyses) seems useful.

Reviewer #3

(Remarks to the Author)

(Remarks on code availability)

The provided code can be used in itself with the provided processed data. The plots and reported statistical test results from the manuscript main text can be reproduced.

The statistics and the corresponding retrospective rating's data in the supplementary are not provided.

Reviewer #4

(Remarks to the Author)

(Remarks on code availability)

I'm not the R user.

I do not think I'm the person who could not review the code properly. However, it seems that they provided the script for the analysis and the relevant data for the reproducibility.

Version 1:

Reviewer comments:

Reviewer #1

(Remarks to the Author)

The revised manuscript addresses all our comments and concerns. The authors have made commendable efforts to substantiate their conclusions, incorporating new analytical pipelines and additional analyses that significantly strengthen and validate their findings.

We have only two comments:

As suggested, the authors have now included correlation analyses between behavioral measures (pain ratings) and neural effects (ROI and NPS/SIIPs), supporting an association between these measures, as well-documented in the literature.

However, to further substantiate the claim that the analgesic effect of nature is driven by changes in nociceptive processing, it would be beneficial for the authors to demonstrate correlations between the individual analgesic effect of nature (e.g., individual deltas in pain ratings) and the corresponding neural responses. I apologize if my earlier comments on this point were not sufficiently clear.

The authors might also give an explanation why the provided MRI acquisition parameters of the fMRI sequence differ from the "classic" association between FOV, matrix size(MS), and voxel size(VS): $FOV=MS*VS$.

Ulrike Bingel and Balint Kincses

(Remarks on code availability)

please see the comments by Balint Kincses.

Reviewer #2

(Remarks to the Author)

We appreciate your substantial efforts in revising the manuscript, including additional analyses of fMRI and behavioral data. We have only minor comments:

1. Some guidelines for interpreting the magnitude of effect sizes, e.g., d_{rm} , would be helpful.
2. Clarification on how standardized signature scores were calculated in Figs. 2D and 2F would also be helpful.
3. The clusters reported in Supplementary Table 3 appear overly large, reducing their utility for identifying regions. It would be helpful if they could provide information about subclusters or peak information.

(Remarks on code availability)

Reviewer #3

(Remarks to the Author)

(Remarks on code availability)

The shared code and data contain all the details to reproduce the reported main and supporting findings. I could reuse the code with the data. While the human readability of the code of the main analysis is good, the human readability of the the supporting_analysis' code can be improved (e.g.: rendered format of the notebook, additional information/text snippets from the manuscript, so the code in itself can be read,...).

Reviewer #4

(Remarks to the Author)

(Remarks on code availability)

Response letter for

Nature exposure induces analgesic effects by acting on nociception-related neural processing

General statement: *We would like to sincerely thank the reviewers for their time and effort in carefully and attentively reading our manuscript. We greatly appreciate the thoughtful and critical evaluation provided, which we believe has significantly contributed to improving the quality of our manuscript. The reviewers' comments encouraged us to re-evaluate relevant aspects of the manuscript and their suggestions motivated us to conduct further analyses that reinforce the robustness and confidence in our findings. We would also like to express our gratitude to the handling editor for acknowledging the relevance of our findings and for the efforts invested in guiding this manuscript through the review process so far. We hope that the revised manuscript now meets the expectations of the reviewers and the editor.*

REVIEWER COMMENTS

Reviewer #1 (Remarks to the Author):

Response to nature exposure induces hypoalgesia by acting on nociception-related neural processing

Reviewer: In this manuscript, Steininger and colleagues examined the impact of nature exposure on the perception and neural processing of pain in a sample of 49 healthy participants. Their findings demonstrate that immersion in a virtual natural environment significantly reduces subjective measures of pain perception, including intensity and unpleasantness. Moreover, the study observed a decreased expression of the Neurologic Pain Signature (NPS)—a well-established multivariate brain signature indicative of pain driven by nociception—under the nature condition compared to control conditions. However, no significant changes were found in the SIIPS1, a signature associated with cognitive-evaluative mechanisms influencing pain perception. This study is both novel and original, addressing a highly relevant topic with significant translational implications. Given that current pharmacological treatments for (chronic) pain conditions are often inadequate or even harmful—particularly in the case of opioids—this research offers valuable insights into alternative therapeutic approaches. The study adheres to the highest standards of open and reproducible science, including preregistration, sharing of processed data, and code sharing, which is highly commendable. I enjoyed reading the introduction and discussion section, which are written very elaborately and comprehensive. However, several parts of the methods and results section could benefit from adding further information. I recommend that the authors address the following points to allow for a more thorough assessment of the validity of the results:

Author response: *We thank the reviewer for the time, effort, and thoughtful critique in carefully examining our manuscript and all related materials. We greatly appreciate the insights provided, which have helped improve the quality of the manuscript. We are also thankful for the positive feedback on our introduction and discussion. In line with the reviewer's recommendations, we have addressed the suggested points in the methods and results sections and ran additional analyses that reinforce the robustness and interpretation of our findings.*

Major points:

1. Reviewer: The authors complement the NPS analysis with pre-registered region-of-interest (ROI) approaches in predefined brain regions (e.g., thalamus, posterior insula). These analyses appear to corroborate the NPS findings, supporting the notion that the analgesic effects of nature interfere with nociceptive and sensory processing rather than higher-level pain processing. However, the analyses in these predefined brain regions are limited to voxels sensitive to the pain vs. non-painful contrast, which introduces circularity with respect to changes in pain due to increased stimulus intensity. For greater transparency, the authors should: a) avoid restricting the ROI analyses to pain vs. non-pain-sensitive voxels, and b) provide the univariate whole-brain maps for the relevant contrasts (nature vs. control conditions) to ensure that significant activity differences are not present outside the ROIs.

Response to reviewer:

a) *We appreciate the reviewer's feedback that caution may be warranted when interpreting the ROI results. Our rationale for focusing on pain-sensitive voxels in the ROI analyses was that the full ROI masks (i.e., including both voxels sensitive and insensitive to pain) would have been rather large, and that including voxels that are non-responsive to the phenomenon of interest (i.e., pain-related processes) could have reduce the sensitivity and the specificity of our analyses. Of note, we determined pain-sensitivity of ROI voxels by using the contrast pain>nopain averaged across (and thus irrespective of) the three conditions of interest (i.e., the three environments). This contrast was, thus, independent to the assessments of condition differences (i.e., comparing pain>nopain independently determined in the three environments). While we hope that clarifying this approach will mitigate any concerns about circularity the reviewer may have, as suggested we have reanalyzed the data using the full ROIs (i.e., without restriction to pain-sensitive voxels) to reinforce the robustness of our results. The results and conclusions remain consistent for the main regions emphasized in our discussion and conclusion sections (i.e., thalamus, S2, and pINS; for details, see below). For the SPL, we observe a shift from a significant main effect of environment to a trend-level effect, while for the aINS, there is a shift from a trend-level effect to a significant main effect of environment. Importantly, these analyses corroborate the robustness of the thalamus, S2 and pINS results across different analytical choices (see also points 8, 9 and 10), and highlight the variability of the SPL and aINS results in response to these choices. We have included this reanalysis in the Supplementary*

Information and reference it in the main text (see lines 274-279 and 679-682 in main text and 133-141, 165-169 and 426-438 in the Supplementary Information).

Main text changes (the text changes also relate to our responses to points 8, 9 and 10 for details on alternative first-level model, outliers, and motion analyses)

274-279

Importantly, the effects in the thalamus, S2, and pINS remained largely consistent after excluding statistical outliers, participants exceeding motion thresholds, when applying an alternative first-level model specification for the MRI data, and when using alternative ROI masks (Supplementary Results). The effects observed in the SPL and aINS were more sensitive to these analytical choices and should, therefore, be interpreted with caution.

679-682

For each ROI, we only included voxels that showed a significant response to painful vs. non-painful stimuli in the pain>no-pain contrast across environments (see Supplementary Results for ROIs encompassing voxels sensitive and insensitive to pain).

Supplementary Information text changes

133-141

Furthermore, to ensure the robustness of our MRI analyses, we conducted several sensitivity analyses across different analytical approaches. Specifically, we reanalyzed the data by: (1) excluding participants identified as statistical outliers, (2) applying a motion threshold of 2 mm, (3) using a more parsimonious first-level model, and (4) employing alternative masks for our ROIs. Detailed descriptions of the results are provided in the Supplementary Results section "Sensitivity analysis". Notably, despite these variations in analytical approaches, results for the NPS and key ROIs, including the thalamus, S2, and pINS remained consistent. However, greater variability in the outcomes was observed for the SPL and aINS, suggesting that findings in these regions should be interpreted with caution.

165-169

Regarding point (4), the main article reports results based on ROIs that include only voxels showing a significant response to painful versus non-painful stimuli in the pain>no-pain contrast across all environments. Following a reviewer's suggestion, we reanalyzed the data using the full ROI masks that encompassed all voxels irrespective of their sensitivity to pain. These additional analyses are detailed in the section "Sensitivity analyses" below.

426-438

Alternative ROIs

The main article reports ROIs that include only pain-sensitive voxels, identified using the pain>no-pain contrast across all environments. We focused on these pain-sensitive voxels due to the large size of the full ROI masks. Nevertheless, we also reran all analyses using the full ROIs, which included voxels both sensitive or insensitive to pain, and present these results here.

Using the full ROIs revealed the same significant main effect of environment in the Thalamus [$F_{(2,48,00)} = 5.94, p = 0.004$], S2 [$F_{(2,48,00)} = 7.73, p = 0.001$], and pINS [$F_{(2,48,00)} = 19.28, p < 0.001$]. However, the previously significant effect shifted to trend-level in the SPL [$F_{(2,48,00)} = 2.89, p = 0.064$], while the trend-level effect in the aINS became significant [$F_{(2,48,00)} = 3.30, p = 0.045$].

Inspection of the previously significant and trend-level planned pairwise comparisons revealed that the results remained consistent for the Thalamus, S2, pINS, aINS, and SPL after correcting for multiple comparisons.

Response to reviewer:

b) Exploring and providing the univariate whole-brain maps is another excellent suggestion. This reveals only one additional area outside the predefined ROIs (at a FWE correction of $p < .05$ at voxel-level), which is the auditory cortex when comparing painful>non-painful shocks in urban vs. nature environments. We propose this indicates differences in the processing of the sounds associated with the respective conditions. While exploratory and thus in need of further validation, this finding adds to the pre-registered analyses by suggesting that urban vs. natural "soundscapes" evoke differences in auditory processing, and that this may have downstream implications for affect, stress, and possibly also pain. We now report this finding in the main text (see lines 279-281 and 478-488) and the Supplementary Information (see lines 346-361).

Main text changes

279-281

Complementary exploratory whole-brain analyses examining activity outside the preregistered ROIs revealed significant voxels only in the primary auditory cortex, when contrasting pain>no-pain in the urban compared to the nature condition (Supplementary Results).

478-488

However, our exploratory whole-brain analysis indicated that, despite being matched in loudness, the urban environment resulted in distinct processing in auditory cortex, compared to nature. This finding provides preliminary evidence that it may not only be the visual quality of the nature stimuli that makes them effective, but that variations in soundscapes may have downstream effects on pain processing as well. Notably, although differences in auditory processing may relate to alterations in pain, they are unlikely to be the primary driver of the observed effects. This is evidenced by the lack of differences in auditory processing between the nature and indoor environments, despite comparable magnitudes of change in pain outcomes across both the urban vs. nature and indoor vs. nature comparison. Thus, further work is needed to explore which specific sensory elements and their combination make natural environments particularly effective in alleviating pain.

Supplementary Information text changes

344-356

Lastly, we conducted exploratory whole-brain analyses to explore putative activation differences outside the preregistered ROIs and signatures. We compared responses to painful>non-painful stimuli across environments (FWE-corrected at voxel level, $p < .05$). In the urban vs. nature comparison, we identified two clusters in the right (peak coordinate [64, -24, 8], cluster size = 588) and left superior temporal gyrus (peak coordinate [-56, -24, 10], cluster size = 426), localized in bilateral auditory cortex and thus suggesting differential effects of urban versus nature soundscapes on auditory processing. The indoor vs. nature comparison revealed a small cluster in the left thalamus (peak coordinate [-4, -18, 2], cluster size = 2), with voxels directly adjacent to those included in our predefined ROI for the thalamus. Beyond the differences in painful processing observed in our signature- and ROI-based confirmatory analyses, the exploratory whole-brain analyses indicate differences in auditory processing when comparing urban and nature environments (note that effects cannot be explained by differences in sound intensity, which had been normalized across urban and nature conditions).

2. Reviewer: Additionally, to further substantiate the claim that the analgesic effects of nature are indeed linked to changes in nociception, the authors should include correlational analyses between individual behavioral analgesic effects and neural responses – both for the ROIs and the NPS.

Response to reviewer:

As suggested by the reviewer, we conducted correlation analyses linking self-reported pain and neural responses to pain. These analyses are now included into the Supplementary Information (see lines 597-619) and briefly referenced in the main text (see lines 294-297). The magnitude of these associations aligns well with previous findings, which report low to moderate correlations between signature and ROI responses with self-reported pain (Han et al., 2022; doi: 10.1016/j.neuroimage.2021.118844 ; Hoeppli et al., 2022; doi: 10.1038/s41467-022-31039-3). Furthermore, this low to moderate association underscores the value of integrating both neural measures and self-reported pain. These approaches provide complementary insights into pain processing (see e.g., Reddan & Wager, 2018; doi: 10.1007/s12264-017-0150-1 ; Woo & Wager, 2016; doi: 10.1097/j.pain.0000000000000442). Importantly, both measures were altered by nature exposure in our study, reinforcing the conclusion that combining these measures enhances our understanding of how pain processing is altered.

Main text changes

294-297

We also performed further exploratory analyses on the association between self-reported and neural pain responses, as well as the role of immersion. Details on these analyses, the results, and their interpretation are documented in the Supplementary Results.

Supplementary Information text changes

597-619

First, we conducted correlation analyses to examine the relationship between self-reported and neural responses to pain. Specifically, we aggregated immediate pain ratings – separated by intensity and unpleasantness – and neural responses for the NPS, SIIPS1, Thalamus, S2, and pINS, separately across environments. Our analyses focused on neural responses that demonstrated robust, significant differences between environments as reported in the “Sensitivity Analyses” section. The correlation analyses revealed moderate positive associations between neural responses and self-reported pain intensity with correlations of $r = .32$ ($p = .024$) for the NPS and $r = .40$ ($p = .009$) for the SIIPS1 response. Similarly, these responses were moderately associated with unpleasantness ratings ($r = .49$, $p = .001$; and $r = .50$, $p = .001$; respectively). All associations between signature responses and ratings were significant after multiple comparison corrections across the four correlations (Bonferroni-Holm). We also observed weak positive, but non-significant correlations between the thalamus ($r = .09$, $p = 1$), S2 ($r = .20$, $p = .51$), and pINS ($r = .05$, $p = 1$) responses and pain intensity. Notably, these regions showed slightly stronger correlations with unpleasantness ratings ($r = .28$, $p = .25$; $r = .33$, $p = .12$; and $r = .24$, $p = .36$; respectively). Applying multiple comparison correction across the six correlations (Bonferroni-Holm) rendered the correlations between ROI responses and unpleasantness ratings as not significant anymore. Thus, higher self-reported pain was significantly associated with higher neural responses in the signature but not the ROI data. The magnitude of these associations is consistent with prior studies, which report weak to moderate associations between self-reported pain and neural responses to pain^{13,22}. These findings highlight the value of integrating both neural and self-reported pain measures, as they offer complementary insights into pain processing. While self-reported pain

reflects the subjective experience, neural data can reveal the underlying brain mechanisms involved in pain perception^{23,24}.

3. Reviewer: The authors report 5 out of 8 of the preregistered hypotheses. This should be made transparent in the manuscript and explained, why relevant information on arousal, immersion etc are not reported. The authors may want to reconsider including the results on the autonomic data which could strengthen the authors interpretation that nature induced analgesic effects are associated with arousal and attention.

Response to reviewer:

We appreciate the reviewer's thorough examination of our pre-registration and the request for transparency in reporting all our hypotheses. The decision to not mention the hypotheses not directly related to pain outcomes was primarily due to space constraints, and our assessment that these findings did not yield results important enough to warrant inclusion in the manuscript. We however acknowledge that such a pragmatic decision is not in the best spirit of the open science credo we followed and endorse and have now included all preregistered hypotheses and their results in the Supplementary Information (see lines 170-177 and 448-591). Additionally, we provide a reference to these findings in the main manuscript to ensure comprehensive reporting (see lines 287-294). In brief, we observed the following results. In line with our prediction, compared to urban or indoor environments, nature was associated with lower negative and higher positive affect. Contrary to our expectations, the effects were stronger for negative than for positive affect. Additionally, in line with our expectations we observed a reduction in pulse rate in the nature condition compared to the urban condition. Lastly, contrary to our expectations, differences in nature connectedness did not moderate the main findings for pain responses.

Main text changes

287-294

Note that in addition to these analyses addressing our hypotheses related to pain outcomes, we had preregistered three additional hypotheses. In brief, we observed (1) that environments significantly differed regarding positive and negative affect (with nature showing higher positive and lower negative affect ratings when compared to urban or indoor settings), (2) that pulse rate was lower in the nature than in the urban setting, and (3) that nature connectedness, contrary to our prediction, did not moderate the main findings (i.e., participants who felt more psychologically close to nature did not show greater benefits compared to those who felt less connected).

Supplementary Information text changes

170-177

Lastly, three additional hypotheses had been part of our preregistration. These hypotheses focused on the impact of the different environments on positive and negative affect, pulse rate, and the potential moderating role of individual differences in nature-connectedness on the effects of nature. The results of these analyses and their interpretations are documented in the Supplementary Results section titled "Additional preregistered analyses". Notably, we observed significant differences in positive and negative affect between the environments, a reduction in pulse rate when comparing the nature and urban environments, and no evidence of moderation effects related to nature connectedness.

448-591

Additional preregistered analyses. As mentioned above, our preregistration included three additional hypotheses related to the impact of the environments on positive and negative affect and pulse rate, as well as the potential moderating role of nature connectedness on subjective and neural pain processing. Due to space constraints and because these hypotheses pertain to effects of nature that are only indirectly related to pain outcomes – placing them outside the core scope of this paper – these results are only briefly reported in the main text but fully documented here.

The effect of environments on positive and negative affect

Based on prior literature¹⁵ we preregistered that nature exposure would lead to heightened positive and reduced negative affect compared to urban or indoor environments. Additionally, we expected these effects to be more pronounced for indicators of positive than for negative affect.

To address these hypotheses, participants completed the Positive and Negative Affect Schedule (PANAS) after each run in the experiment¹⁶. The PANAS measures subjective levels of positive and negative affect using 10 items each. Each item represents a specific feeling, and participants indicate on a scale from 1 ("very slightly or not at all") to 5 ("extremely") how much the respective feeling applies to them at that moment. The two subscales of the PANAS – positive and negative affect – were calculated by averaging the items corresponding to each type of affect valence. To assess whether positive affect (PA) or negative affect (NA) differed across the three environments, we compared the PANAS ratings after each pain run, which included the presentation of painful and non-painful shocks alongside the environment. We calculated a LMM using affect (PA and NA) as the dependent variable, to be predicted by the fixed effect of environment (nature as a reference), valence (NA as a reference), and their interaction (with random slopes and intercepts for environment, valence, and their interaction by participant).

We observed a significant main effect of environment [$F_{(2,111.58)} = 9.09, p < 0.001$], valence [$F_{(2,48.00)} = 114.36, p < 0.001$], and their interaction [$F_{(2,49.02)} = 16.63, p < 0.001$]. In line with our predictions, planned pairwise

comparisons revealed significantly lower negative affect in nature compared to urban [$b = -0.33$, $SE = 0.06$, $t_{(48)} = -5.18$, $p < 0.001$ one-tailed, 95% $CI = [-0.463, -0.204]$, $d_{rm} = -0.76$] or nature compared to indoor environments [$b = -0.21$, $SE = 0.06$, $t_{(48)} = -3.60$, $p < 0.001$ one-tailed, 95% $CI = [-0.322, -0.092]$, $d_{rm} = -0.58$]. Additionally, positive affect was significantly higher in nature compared to urban [$b = 0.14$, $SE = 0.07$, $t_{(48)} = 1.98$, $p = 0.026$ one-tailed, 95% $CI = [-0.002, 0.276]$, $d_{rm} = 0.21$] and nature compared to indoor environments [$b = 0.34$, $SE = 0.07$, $t_{(48)} = 4.78$, $p < 0.001$ one-tailed, 95% $CI = [0.195, 0.478]$, $d_{rm} = 0.55$]. We also found significantly higher positive affect [$b = 0.20$, $SE = 0.07$, $t_{(48)} = 2.80$, $p = 0.007$, 95% $CI = [0.056, 0.343]$, $d_{rm} = 0.33$], and higher negative affect [$b = 0.13$, $SE = 0.06$, $t_{(48)} = 2.02$, $p = 0.049$, 95% $CI = [0.001, 0.253]$, $d_{rm} = 0.28$] for urban compared to indoor environments.

Thus, in line with our prediction, receiving painful stimulation while being exposed to nature was associated with higher positive and lower negative affect compared to exposure to urban or indoor environments. Contrary to our prediction, the effect was more pronounced for negative (moderate to high effect size), than for positive affect (low to moderate effect size). Since our study, unlike most previous nature-based research, investigated affective processing in an aversive context (i.e., painful electrical stimulation), we speculate that this may have resulted in greater variability in negative affective responses, allowing for more pronounced differences across environments. Furthermore, we also observed differences when comparing urban and indoor environments, although these effects were generally smaller and less consistent in terms of their direction. Although the observed differences in self-reported affect are consistent with previous findings and appear to support stress recovery theory (SRT), the changes in neural pain outcomes do not suggest that the reduction in pain was driven by these affective changes (see discussion in main text).

The effect of environments on pulse rate

We preregistered that pulse rate will be decreased during exposure to nature as compared to urban environments. Since we were not aware of studies comparing differences in pulse rate for nature and indoor environments, we restricted the predictions to the nature vs. urban comparison.

To investigate this hypothesis, we recorded pulse rate using the built-in peripheral pulse unit of the MRI scanner (PPU, Siemens Medical, Erlangen, Germany). The raw photoplethysmography (PPG) signal, sampled at 200 Hz, was processed for each functional run. Pulse rate was derived from the intervals between pulse peaks and is expressed in beats per minute (BPM). Signal processing and RR interval detection were performed using the Python Heart Rate Analysis Toolkit "heartpy"¹⁷. Each participant's data were manually inspected to ensure accurate peak detection and to identify any missed or incorrectly detected peaks. One participant was excluded from the analysis due to an erroneous raw PPG signal, which could not be processed. We calculated a LMM using BPM as the dependent variable, to be predicted by the fixed effect of environment (nature as a reference, with random intercepts for environment by participant).

We observed no significant main effect of environment. However, since we were interested in the difference between nature and urban environments, we continued to perform a planned pairwise comparison. This comparison revealed significantly lower BPM for the nature environment when compared to the urban environment, with a small effect size [$b = -0.86$, $SE = 0.42$, $t_{(94)} = -2.04$, $p = .043$, 95% $CI = [-1.700, -0.025]$, $d_{rm} = -0.09$]. Exploratory post-hoc pairwise comparisons revealed no difference when comparing the nature and the indoor environment [$b = -0.42$, $SE = 0.42$, $t_{(94)} = -0.99$, $p = 0.321$, 95% $CI = [-1.260, 0.417]$, $d_{rm} = -0.05$] or the urban and indoor environment [$b = 0.44$, $SE = 0.42$, $t_{(94)} = 1.05$, $p = 0.298$, 95% $CI = [-0.397, 1.280]$, $d_{rm} = 0.05$]. Thus, in line with our prediction, pulse rate was significantly lower when comparing nature and urban environments. However, the effect size was very small and translated to a change of approximately one BPM between nature ($M = 65.09$, $SD = 9.05$) and urban environments ($M = 65.96$, $SD = 9.44$). Moreover, the lack of differences to the neutral indoor environment makes it difficult to interpret the underlying mechanism of these differences.

The effect of nature connectedness on the impact of environment

Drawing on existing literature that demonstrated the beneficial effects of nature are influenced by individual levels of nature connectedness¹⁸, we preregistered our hypothesis that participants with a stronger connection to nature will exhibit greater changes in pain responses when exposed to nature.

To investigate our hypothesis, we used the Nature Connection Index (NCI) to assess individual levels of nature connectedness¹⁹. The NCI comprises six items (e.g., "I always find beauty in nature") rated on a 7-point scale, ranging from "completely disagree" to "completely agree". Following the authors' recommendations, we calculated overall levels of nature connectedness by using a weighted point index, where higher scores indicate greater nature connectedness. In the subsequent analyses, we recalculated all models that included significant main or interaction effects of the variable environment on immediate self-reported or neural data. We included nature connectedness (centered) and its interaction with the other main effects as independent variables. For the ROI data, we focused the analyses on the thalamus, S2 and pINS, as these regions were characterized by robust effects of environment across different analytical choices (see "Sensitivity analyses" above).

For the immediate pain ratings, we employed a LMM with ratings as the dependent variable, to be predicted by the fixed effect of environment (nature as a reference), rating content (intensity as a reference), nature connectedness and their interactions (with random intercepts and slopes for environment, rating content, and their interactions). For the signature data, we recalculated a LMM using signature responses as the dependent variable, to be predicted by the effect of environment (nature as a reference), signature (NPS as a reference), nature connectedness and their interactions (with random intercepts and slopes for environment and signature). Furthermore, for the thalamus, S2, and pINS, we recalculated a LMM using the ROI response as the dependent

variable, to be predicted by the fixed effect of environment (nature as a reference), hemisphere (left hemisphere as a reference), nature connectedness and their interactions (with random intercepts and slopes for environment and hemisphere). We calculated likelihood ratio tests (LRT) to compare the performance of two models per dependent variable: one model including nature connectedness as a moderator variable and one excluding it. A significant result from the LRT indicates that the more complex model, which incorporates nature connectedness as a fixed effect, provides a significantly better fit to the data compared to the simpler model. Additionally, we checked the models for significant main effects of nature connectedness or its interaction with the variable environment.

The results of the LRTs are summarized in Supplementary Table 17. We did not observe significant improvements in model fit when incorporating nature connectedness as a fixed effect into our models. Furthermore, neither a significant main effect of nature connectedness nor any significant interactions with the variable environment were detected. Note, that the LRT for the thalamus responses approached trend-level significance. However, the main effect of nature connectedness, as well as its two-way and three-way interactions with environment, did not reach significance (all $p > .34$).

Thus, contrary to our preregistered hypothesis, our data did not reveal that nature connectedness significantly altered the impact of nature on self-reported and neural responses to pain. This contrasts with the literature suggesting that the beneficial effects of nature are moderated by nature connectedness¹⁸, a discrepancy that may arise from differences in outcome variables or the type of nature exposure investigated. Previous studies demonstrating this moderation effect focused on measures of mental health and well-being and examined the effect of real-world nature exposure. It is possible that the effects on mental health and well-being, which are less tangible than immediate pain experiences in response to a specific stimulus, may be more pronounced. Furthermore, virtual nature may not provide the same increased benefits for individuals with higher levels of nature connectedness, as it could differ significantly from their typical experiences in nature. However, employing non-real, virtual nature rather than real-world stimuli also minimized potential confounding influences stemming from familiarity and prior experiences with specific settings. Notably, such prior experiences and learned associations may affect the salutary effects of nature. It has been suggested that positive past interactions can shape the restorative effects of nature through conditioning processes²⁰. When positive emotions are repeatedly linked with specific environments, cues from these environments may eventually elicit similar positive responses. Additionally, positive past experiences are related to high levels of nature connectedness which in turn are associated with greater perceived benefits from nature²¹. Since we found no moderating effect of nature connectedness – an outcome typically associated with positive prior experiences - on the pain outcomes, our current findings do not appear to support these suggestions. However, these interpretations are speculative, especially as nature connectedness is only an indirect measure of positive past experiences. Thus, future studies should investigate more directly the influence of prior experiences and learning mechanisms on the pain-relieving effects of nature.

752-754

Supplementary Table 17. Likelihood ratio tests (LRT) comparing models with or without nature connectedness to explain self-reported and neural responses to pain.

Variable	Model	npar	AIC	BIC	χ^2	df	p
Immediate ratings	Excl. NCI	28	5449.9	5603.2	5.67	6	0.461
	Incl. NCI	34	5456.3	5642.4			
Signature Response	Excl. NCI	17	761.9	824.6	6.26	6	0.395
	Incl. NCI	23	767.7	852.4			
Thalamus Response	Excl. NCI	17	410.7	473.4	12.24	6	0.057
	Incl. NCI	23	410.5	495.2			
S2 Response	Excl. NCI	17	621.7	684.4	7.13	6	0.308
	Incl. NCI	23	626.6	711.3			
pINS Response	Excl. NCI	17	692.9	755.6	8.25	6	0.219
	Incl. NCI	23	696.7	781.4			

Note: AIC = Akaike Information Criterion, BIC = Bayesian Information Criterion, df = degrees of freedom, NCI = nature connectedness index, npar = number of parameters in model, pINS = posterior insula, S2 = secondary somatosensory cortex.

Other points:

4. Reviewer: Introduction:

In the introduction mentions reporting bias as a potential confounder in prior research. Could the authors elaborate on how this issue was addressed in their design (e.g., instructions given to participants)? If this was not addressed, it should be acknowledged as a limitation.

Response to reviewer:

We would like to clarify that we did not specifically address reporting biases through our instructions. Rather, we wanted to make the multidisciplinary audience of Nature Communications aware that such biases exist, and that our approach to combine neuroimaging alongside self-report is thus able to address such limitations. Naturally,

our results based on self-report suffer from similar shortcomings. We thank the reviewer for urging us to make this more explicit, and now acknowledge it in our limitations section (lines 489-494).

Main text changes
489-494

Third, while we highlight the limitations of subjective pain measures in previous studies due to potential reporting biases, our own self-reported data faced similar shortcomings. That said, the convergence between neuroimaging and self-report findings instills added confidence that the present and prior findings are not predominantly driven by such biases. Additionally, since the neural results contrast with those from placebo studies^{30,36}, it seems less likely that they are entirely attributable to expectation or demand characteristics.

5. Reviewer: Methods:

Matching Procedure: The authors emphasize that previous studies on the effects of nature exposure often used suboptimal designs and highlight the importance of incorporating a well-matched control condition. They also claim that the videos used in their study were "carefully matched." Could the authors provide a more detailed explanation of the matching procedure, including the criteria for matching (e.g., subjective measures, physical properties)? Additionally, please specify whether the images were sourced from previously published studies.

Response to reviewer:

In line with the reviewer's suggestion, we have included a more detailed explanation of the matching process and a more nuanced description of the virtual environments in the methods section of the revised manuscript (see lines 550-560). Together with the text included in the previous manuscript it highlights the criteria used for matching and clarifies in more detail that the stimuli were indeed sourced from previously published studies.

Main text changes
550-560

The natural and urban environments were adapted from previously published studies^{34,42}. They were closely matched in terms of various visual characteristics, structural proportions, and physical features such as lighting conditions and loudness of sounds. Both environments were designed to be generally favorable by including and matching elements that prior research has identified as appealing, such as a large water body, reflective surfaces, green foliage, and a high level of complexity and openness of space⁴⁴⁻⁴⁶. Specifically, the natural setting was created first and included a large central lake (with observable wind ripples), trees by the side of the lake (with rustling leaves), and an animation showing the shifting position of the sun and cloud movements. The urban condition was constructed by adding human-made elements to this basic scene, including buildings on the far side of the lake, a paved path, a short wall, and benches on the nearside of the lake.

6. Reviewer: The author mention in the main text a video and a pain block, however, no information about the video block (condition order, timing). The preregistration details it, but it would be beneficial to the reader to mention in the main text as well. Were the scripts, to enhance immersion, read before each condition? How much pause were between runs? How long was the total measurement?

Response to reviewer:

We thank the reviewer for highlighting the necessity to include additional information that may be relevant to readers. In accordance with the suggestion, we have included more details in the Methods sections (see lines 538-546 and 572-578). Additionally, we clarified information regarding the immersion scripts and the overall timing of the experiment. This addition enhances clarity and completeness in the respective sections.

Main text changes
538-546

Afterward, they entered the MRI scanner and were alternately exposed to blocks of virtual stimuli each depicting a different environment (5 min), directly followed by blocks showing the same environment accompanied by electrical shocks (9 min; from here on referred to as "video" and "pain" blocks, respectively). This design enabled participants to familiarize themselves with each respective environment before its presentation alongside the pain stimuli. The order of the presented environments was counterbalanced across participants. Participants completed several ratings after each block, leading to an approximately 5 min pause between them. The total duration of the study procedure was 120 min.

572-578

During the video blocks, participants observed each scene while being instructed to imagine themselves being present in the specific environment. This was facilitated by reading through a short script immediately preceding each video block. The scripts were based on previous nature-based guided imagery interventions⁴⁸. During pain blocks, participants were instructed to read a short immersion script, then re-watched the same environment from the preceding video block while receiving electrical shocks, with the video playing in the background.

7. Reviewer: The authors refer to their previous work regarding the pain calibration procedure. At least brief information on this and where it was performed (in/outside scanner) should be added to the supplement.

Proposed response to reviewer:

We thank the reviewer for highlighting the necessity to include additional information that may be relevant to readers. We added a brief description of the pain calibration procedure in the Methods section (see lines 583-589).

Main text changes

583-589

The calibration consisted of three phases, separated by breaks of approximately 3 min, and was conducted inside the scanner. Participants received an initial low-intensity shock (0.05 mA) in the first two phases, followed by progressively stronger shocks rated on a scale from 0 to 8. Each phase concluded when a shock was rated as 8, resulting in a variable number of shocks for each participant. In the third phase, shocks reflecting the average intensities of 1 and 6 from the first two phases were pseudorandomly administered and rated by the participants to confirm the stability of their perceived intensity.

8. Reviewer: fMRI Modelling: Please specify which regressors were included in the first-level general linear model (GLM), such as anticipation, stimulation, rating, etc.

Response to reviewer:

We thank the reviewer for highlighting that the details of our first-level analysis need clarification. Our original approach prioritized model parsimony, which is why we chose not to include additional event-related regressors beyond the delivery of painful and non-painful shocks to the design matrix, to avoid 'overfitting' the data. While the reviewer did not expressly criticize this approach, the comment prompted us to reconsider our scientific rationale. Consequently, we decided to assess the robustness of our findings by re-running all analyses using a first-level model that also modeled cue (for painful and non-painful stimuli separately) and rating as additional regressors. In brief, this analysis yielded results that were virtually identical to those resulting from the use of the more parsimonious model, providing converging evidence in the sense of a multiverse analysis (Botvinik-Nezer et al., 2020, doi: 10.1038/s41586-020-2314-9). In more detail, we first assessed whether the more comprehensive model led to a more precise and robust identification of the areas involved in pain processing, using the contrast pain>no-pain (pooled across conditions, i.e. using a contrast that is orthogonal to and unrelated to our main environment-based hypotheses and thus would not introduce potential bias or circularity re: condition effects). As this was indeed the case (see Supplementary Information 230-265), we then continued to perform our pre-registered hypotheses testing condition effects with the more comprehensive first-level model. These analyses confirmed the same significant interaction effect between environment*signature for the signature data, as well as the same main effects of environment in the thalamus, S2 and, pINS. Additionally, planned pairwise comparisons revealed the same significant differences as in the original analyses. Apart from the excellent convergence and virtual identity of findings, the following arguments speak for using the more comprehensive model as the main analysis approach. In the more parsimonious model, the implicit baseline is a mix of non-modeled events; thus, including these events as regressors, the baseline becomes more readily interpretable. Moreover, upon re-examining the literature employing comparable study designs, we found this approach to be more prevalent (e.g. Botvinik-Netzer et al., 2024, doi: 10.1038/s41467-024-50103-8; Wielgosz et al., 2022, doi: 10.1176/appi.ajp.21020145). Nevertheless, we prefer to report both analyses and their findings, with the updated and original analysis included in the main text (see lines 238-240, 274-277, and 644-646) and Supplementary Information (see lines 133-141, 155-164, and 401-424), respectively. This ensures a fully transparent documentation of all analyses performed, communicates the adapted rationale for using the more comprehensive first-level model as the primary one in the end, and by way of the "multiverse" approach with full converging evidence strengthens our conclusions.

Main text changes

238-240

Importantly, these effects remained largely consistent after excluding statistical outliers, participants exceeding motion thresholds, and when applying an alternative first-level model specification for the MRI data (Supplementary Results).

274-277

Importantly, the effects in the thalamus, S2, and pINS remained largely consistent after excluding statistical outliers, participants exceeding motion thresholds, when applying an alternative first-level model specification for the MRI data, and when using alternative ROI masks (Supplementary Results).

644-646

A design matrix was specified with the following five experimental regressors per environment (i.e., run): anticipation of painful shocks, anticipation of non-painful shocks, delivery of painful shocks, delivery of non-painful shocks, and rating.

Supplementary Information text changes

133-141

Furthermore, to ensure the robustness of our MRI analyses, we conducted several sensitivity analyses across different analytical approaches. Specifically, we reanalyzed the data by: (1) excluding participants identified as statistical outliers, (2) applying a motion threshold of 2 mm, (3) using a more parsimonious first-level model, and

(4) employing alternative masks for our ROIs. Detailed descriptions of the results are provided in the Supplementary Results section “Sensitivity analysis”. Notably, despite these variations in analytical approaches, results for the NPS and key ROIs, including the thalamus, S2, and pINS remained consistent. However, greater variability in the outcomes was observed for the SPL and aINS, suggesting that findings in these regions should be interpreted with caution.

155-164

Regarding point (3), we initially employed a parsimonious first-level model to reduce the risk of overfitting. This model included eight regressors: delivery of painful shocks, delivery of non-painful shocks, and six motion regressors. This approach was also outlined and reported in the preprint version of this article. Following reviewer feedback and in alignment with standard modeling approaches used in similar research^{12,14}, we revised the first-level model to incorporate three additional regressors. This revised more comprehensive model includes the following 11 regressors: anticipation of painful shocks, anticipation of non-painful shocks, delivery of painful shocks, delivery of non-painful shocks, ratings, and six motion regressors. This revised approach is reported in the main article, while the initial parsimonious model is detailed in the section “Sensitivity analyses” below.

401-424

Parsimonious first-level model

We repeated all analyses using an additional first-level model for the MRI data. Our original approach prioritized a parsimonious first-level model, which included 8 regressors for each environment: one regressor for delivery of painful shocks, one regressor for delivery of non-painful shocks, and six nuisance regressors accounting for motion. This approach was initially presented in the preprint of this article. After receiving feedback from the reviewers, re-examining the literature on similar research questions and designs^{12,14}, and principled arguments regarding the interpretability of the used implicit baseline, we reconsidered this approach and implemented an additional, more comprehensive first-level model. This model included 11 regressors for each environment: one regressor for the anticipation of painful shocks, one for the anticipation of non-painful shocks, one for delivery of painful shocks, one for delivery of non-painful shocks, one for the rating phase, and six nuisance regressors accounting for motion. The findings using this more comprehensive model, following validation checks orthogonal to the main hypotheses (see main text), are now reported in the main text. For transparency and completeness, we also present the results of the more parsimonious first-level model here.

Using the more parsimonious model yielded the same significant interaction between environment*signature [$F_{(2,96.00)} = 6.04, p = 0.003$] for the signature data. Additionally, we observed the same significant main effects of environment in the Thalamus [$F_{(2,47.99)} = 5.53, p = 0.006$], S2 [$F_{(2,48.00)} = 5.16, p = 0.009$], and pINS [$F_{(2,48.00)} = 9.28, p = 0.0003$]. However, the previously significant and trend-level effects in the SPL [$F_{(2,48.00)} = 2.08, p = 0.136$] and aINS [$F_{(2,48.00)} = 2.39, p = 0.101$] were no longer observed.

Inspection of the planned pairwise comparisons for the significant main and interaction effects revealed the same overall results. There was no change in significance regarding the planned pairwise comparisons in the NPS, Thalamus, S2, and pINS.

9. Reviewer: In the preregistration, outliers are defined as values higher than 3 standard deviations from the mean. Based on Fig2D,F, it seems that the data of the NPS and SIIPS1 still contain outliers. Please specify why the authors deviated from the preregistered criteria and provide complementary analyses with these exclusions.

Response to reviewer:

We initially planned to exclude outliers but decided to deviate from this, as outliers may represent meaningful individual variability. Exclusion could also reduce ecological validity, particularly when investigating complex neural markers. For instance, for the multivariate patterns used in this study (NPS or SIIPS1) it is common to encounter values far from the mean that may be identified as statistical outliers. Upon reviewing the distributions of these markers in prior research and noting similar values we decided to include these observations in our analysis. For reference see: Figure S1 in Han et al. (2022, doi: 10.1016/j.neuroimage.2021.118844) or Figure S2 and S4 in Botvinik-Netzer et al. (2024, doi: 10.1038/s41586-020-2314-9). However, prompted by the reviewer’s comment, we repeated all analyses excluding participants with values greater or lower than 3 standard deviations from the mean. This supporting analysis is now provided in the Supplementary Information. Crucially, excluding outliers did not affect the significance of fixed effects in the linear mixed models, and we do not find relevant changes in main effects in the signature data, the thalamus, the S2, and the pINS when excluding statistical outliers. However, some planned pairwise comparisons shifted from significant to trend level. Again, in the spirit of a multiverse approach, we document both analyses and their overall convergence in the main text (lines 238-240 and 274-277) and Supplementary Information (lines 133-141, 142-154 and 361-381).

Main text change

238-240

Importantly, these effects remained largely consistent after excluding statistical outliers, participants exceeding motion thresholds, and when applying an alternative first-level model specification for the MRI data (Supplementary Results).

274-277

Importantly, the effects in the thalamus, S2, and pINS remained largely consistent after excluding statistical outliers, participants exceeding motion thresholds, when applying an alternative first-level model specification for the MRI data, and when using alternative ROI masks (Supplementary Results).

Supplementary Information text change

133-141

Furthermore, to ensure the robustness of our MRI analyses, we conducted several sensitivity analyses across different analytical approaches. Specifically, we reanalyzed the data by: (1) excluding participants identified as statistical outliers, (2) applying a motion threshold of 2 mm, (3) using a more parsimonious first-level model, and (4) employing alternative masks for our ROIs. Detailed descriptions of the results are provided in the Supplementary Results section "Sensitivity analysis". Notably, despite these variations in analytical approaches, results for the NPS and key ROIs, including the thalamus, S2, and pINS remained consistent. However, greater variability in the outcomes was observed for the SPL and aINS, suggesting that findings in these regions should be interpreted with caution.

142-154

Regarding points (1) and (2), the primary analyses presented in the main manuscript do not exclude statistical outliers or participants exceeding motion thresholds. Although we initially planned to exclude these cases, we reconsidered this approach (before performing any analyses or looking at any data) because outliers may capture meaningful individual variability and exclusions could compromise ecological validity. Moreover, residual movement was modeled as a nuisance regressor, and visual inspection of the data (orthogonal to any hypotheses, pain>no-pain) revealed that participants exceeding the 2mm movement criterion showed no major signs of movement artifacts. These considerations are particularly pertinent when analyzing multivariate patterns such as the NPS and SIIPS1. Upon reviewing the distribution of these pattern responses in prior research^{12,13}, we concluded that deviating values represent valid variability within our sample. Consequently, we opted to retain these data in our analysis. However, for full transparency and completeness of reporting all analyses, we report the results of analyses excluding these participants in the section "Sensitivity analyses" below.

361-381

Exclusion of statistical outliers

We repeated all analyses reported in the main manuscript, excluding statistical outliers defined as values exceeding 3 standard deviations from the mean.

Applying this criterion led to the exclusion of two participants for the self-reported immediate pain ratings, four participants for the signature response data, and the following exclusions for the extracted ROIs: one participant for the Thalamus, two for S2, three for the pINS, two for the amygdala, one for the S1, six for the SPL, and three for the aINS.

Excluding these participants did not alter the main effect of environment [$F_{(2,46,12)} = 12.98, p < 0.001$] or the significant interaction between environment*rating type [$F_{(2,73,19)} = 8.51, p < 0.001$] for the immediate ratings. Similarly, the significant interaction effect between environment*signature [$F_{(2,88,91)} = 5.36, p = 0.006$] remained unchanged. For the ROIs we observed a significant main effect of environment in the Thalamus [$F_{(2,46,99)} = 7.45, p = 0.0015$], S2 [$F_{(2,45,99)} = 6.29, p = 0.0038$], and pINS [$F_{(2,44,99)} = 17.61, p < 0.001$]. However, the previously observed significant and trend-level effects for the SPL [$F_{(2,42,00)} = 1.86, p = 0.168$], and aINS [$F_{(2,45,00)} = 2.12, p = 0.133$] changed and were no longer significant.

Inspection of the planned pairwise comparisons for the significant main and interaction effects revealed similar overall results. Most significant planned pairwise comparisons remained significant after multiple comparison correction, with two exceptions. The comparison between nature and indoor environments shifted from significant to trend level in the NPS [$b = -0.23, SE = 0.16, t_{(82,3)} = -1.42, p = .079$ one-tailed, 95% CI = [-0.547, 0.091]] and the pINS [$b = -0.41, SE = 0.19, t_{(45)} = -2.09, p = .063$ one-tailed, 95% CI = [-0.795, -0.015]].

10. Reviewer: Were there any participants who were excluded based on motion criteria? What was the exact motion criteria?

Response to reviewer:

In our analysis, we did not exclude participants based on motion criteria, as we modeled residual movement as a nuisance regressor, and visual inspection of the data revealed that participants exceeding the 2mm movement criterion showed no major signs of movement artifacts. In response to the reviewer's comment, we have now included a re-analysis applying a 2-mm threshold for excessive head movement. The results of this analysis have been added to the Supplementary Information alongside the other sensitivity checks reported above (see statistical outliers, changed ROI definition, adapted first-level GLM). Applying the threshold led to the exclusion of 9 participants, resulting in a final sample of $n = 40$. Importantly, the exclusion of these participants does not alter the significance of the fixed effects in our LMM for the signature data and the key regions of interest, including the S2, pINS and thalamus. This re-analysis further highlights that the key regions (thalamus, S2 and pINS) and the NPS response, emphasized in the original manuscript, remain largely unaffected by the variation in analysis choices. In our view, this further underscores the robustness of our findings. The results are reported in the main text (lines 238-240 and 274-277) and Supplementary Information (lines 133-141, 142-154 and 383-399).

Main text change

238-240

Importantly, these effects remained largely consistent after excluding statistical outliers, participants exceeding motion thresholds, and when applying an alternative first-level model specification for the MRI data (Supplementary Results).

274-277

Importantly, the effects in the thalamus, S2, and pINS remained largely consistent after excluding statistical outliers, participants exceeding motion thresholds, when applying an alternative first-level model specification for the MRI data, and when using alternative ROI masks (Supplementary Results).

Supplementary Information text change

133-141

Furthermore, to ensure the robustness of our MRI analyses, we conducted several sensitivity analyses across different analytical approaches. Specifically, we reanalyzed the data by: (1) excluding participants identified as statistical outliers, (2) applying a motion threshold of 2 mm, (3) using a more parsimonious first-level model, and (4) employing alternative masks for our ROIs. Detailed descriptions of the results are provided in the Supplementary Results section "Sensitivity analysis". Notably, despite these variations in analytical approaches, results for the NPS and key ROIs, including the thalamus, S2, and pINS remained consistent. However, greater variability in the outcomes was observed for the SPL and aINS, suggesting that findings in these regions should be interpreted with caution.

142-154

Regarding points (1) and (2), the primary analyses presented in the main manuscript do not exclude statistical outliers or participants exceeding motion thresholds. Although we initially planned to exclude these cases, we reconsidered this approach (before performing any analyses or looking at any data) because outliers may capture meaningful individual variability and exclusions could compromise ecological validity. Moreover, residual movement was modeled as a nuisance regressor, and visual inspection of the data (orthogonal to any hypotheses, pain>no-pain) revealed that participants exceeding the 2mm movement criterion showed no major signs of movement artifacts. These considerations are particularly pertinent when analyzing multivariate patterns such as the NPS and SIIPS1. Upon reviewing the distribution of these pattern responses in prior research^{12,13}, we concluded that deviating values represent valid variability within our sample. Consequently, we opted to retain these data in our analysis. However, for full transparency and completeness of reporting all analyses, we report the results of analyses excluding these participants in the section "Sensitivity analyses" below.

383-399

Exclusion of participants above motion threshold

We repeated all analyses reported in the main manuscript, excluding participants with head motion exceeding a threshold of 2 mm. This led to the exclusion of 9 participants, resulting in a final sample size of $n = 40$.

Excluding these participants did not affect the significant interaction between environment*signature [$F_{(2,116.99)} = 5.89, p = 0.004$] for the signature data. Regarding the ROIs, we observed a significant main effect of environment in the Thalamus [$F_{(2,39.00)} = 6.06, p = 0.005$], S2 [$F_{(2,39.00)} = 7.35, p = 0.0019$], and pINS [$F_{(2,39.00)} = 15.03, p < 0.001$]. However, the previously significant effect in the SPL shifted from significant to trend-level [$F_{(2,39.00)} = 3.04, p = 0.058$], and the previously observed trend for the aINS was no longer present [$F_{(2,39.00)} = 1.36, p = 0.266$].

Inspection of the planned pairwise comparisons for the significant main and interaction effects revealed similar overall results. Except for the following comparisons, all significant planned pairwise comparisons remained significant after multiple comparison corrections. The significant difference between nature and indoor environments in the pINS shifted to trend-level [$b = -0.37, SE = 0.21, t_{(39)} = -1.78, p = .053$ one-tailed, 95% CI = [-0.780, 0.049]]. Similarly, the previous trend-level effect in the SPL [$b = -0.51, SE = 0.32, t_{(39)} = -1.63, p = .11$ one-tailed, 95% CI = [-1.150, 0.125]] disappeared when comparing nature and urban environments.

11. Reviewer: Please specify the used fMRI sequence (e.g.: 2D GRE EPI) Please double check the provided parameters (there is a mismatch between the fMRI's FOV, matrix size, and voxel size). Specify the sequence for field map acquisition. How was the field map correction checked?

Response to reviewer:

We thank the reviewer for the careful inspection of our reported parameters. Our functional scans had the following parameters: matrix size: 96 x 96 x 36, voxel size: 2.18 x 2.18 x 3.84 mm³; the slice thickness was 3.50mm, with an interslice gap of 0.34mm. Note that in the revised manuscript we have reported these MRI scanning parameters with two decimal places, to reflect our scanning protocol more precisely. We have updated the information in the manuscript accordingly and hope the information is now sufficiently clear (see lines 619-624). In addition, we now provide detailed information regarding the field map acquisition sequence and how correction was checked (see lines 628-635).

Main text change

619-624

Each run acquired a separate functional volume using a multiband-accelerated gradient echo echoplanar imaging sequence, for one of the three pain blocks using the following parameters: Repetition time (TR) = 800 ms, echo

time (TE) = 34 ms, flip angle = 50°, field of view (FOV) = 138 mm, multi-band acceleration factor = 4, interleaved multi-slice mode, interleaved acquisition, matrix size = 96 × 96 × 36, voxel size = 2.18 × 2.18 × 3.84 mm³, 36 axial slices of the whole brain with slice thickness = 3.50 mm and an interslice gap of 0.34 mm.

628-635

Furthermore, field map images were acquired using a dual-echo gradient echo sequence to correct the functional images for magnetic field inhomogeneities, with the following parameters: TR = 400 ms, TE1 = 4.92 ms, TE2 = 7.38 ms, flip angle = 60°, FOV = 138 mm, matrix size = 128 × 128 × 36, voxel size = 1.72 × 1.72 × 3.84 mm³, 36 axial slices aligned with the orientation of the functional images, and slice thickness = 3.84 mm. Field map correction was checked visually by comparing corrected and uncorrected images and inspecting potential distortions in areas that are prone to artifacts (e.g., near the sinuses, edges of the brain).

12. Reviewer: Results

One of the behavioural ratings (intensity) has a value of 4.2. (Fig2A, indoor condition). As the rating was measured in an ordinal scale, this is not plausible. Please systematically check all data and correct.

Response to Reviewer

We thank the reviewer for this attentive comment and the careful inspection of the data and graphs. The pain intensity rating of 4.2 is not an error but arises from a single missing value among the total of 1764 ratings across all participants and environments. This value was imputed using the mean of the other five ratings from the same environment (indoor) for the respective participant, which resulted in a value of 4.2. This imputation was documented in the original R script (line 38), but we did not consider it important enough to warrant inclusion in the manuscript. To avoid potential confusion for the readership we now report the repeated analysis without imputing this value (treating it as missing). The results are virtually identical to the original analysis (see below). We updated the statistical details in the text and figure in the manuscript which is now based on the data without imputation.

Fixed Effect	With imputation	Without imputation
Environment	$[F_{(2,48.14)} = 12.48, p < 0.001]$	$[F_{(2,48.14)} = 12.49, p < 0.001]$
Rating content	$[F_{(1,48.00)} = 17.51, p < 0.001]$	$[F_{(1,48.00)} = 17.52, p < 0.001]$
Environment*Content	$[F_{(2,81.14)} = 9.19, p < 0.001]$	$[F_{(2,81.11)} = 9.19, p < 0.001]$

13. Reviewer: LI 170 and follows: Modelling of the behavior data resulted in a significant interaction effect. The current phrasing of the meaning of this interaction is a bit confusing (“interaction reflected that the magnitude, but not the overall pattern”). Please clarify.

Response to reviewer:

We thank the reviewer for highlighting the necessity of changing the phrasing to ensure clarity. We changed the phrasing of the interaction effect of the behavioral data and hope that it is now sufficiently clear (see lines 186-189).

Main text change

186–189

Investigations of the beta parameters and planned pairwise contrasts suggested that while both types of ratings were lower in the nature environment compared to the urban or indoor settings, the magnitude of change differed between unpleasantness and intensity ratings.

14. Reviewer: Effect size calculation: In the calculation of the reported effect sizes, the average within subject variance in each condition was used instead of the within condition variance. This may lead to an overestimation of the reported effect size of the behavioural outcomes (intensity and unpleasantness). The inference made from these two estimates would not change, but for completeness I would recommend to correct it. Additionally, the calculated effect sizes estimated from a simple repeated measure (basically a paired t test). There is no consensus, how to report standardized effect sizes from mixed model, therefore, the authors approach can be accepted. However, it would raise the necessity of the use of a (complex) mixed model, as the main conclusion are based on preregistered directed tests.

Response to reviewer:

We thank the reviewer for this expert feedback and the constructive suggestions on how to deal with the complexity of effect size estimates. In line with the recommendation, we corrected the calculated effect size for the behavioral outcomes of immediate pain ratings and updated this in the manuscript (see lines 181-185 and 192-201). As correctly outlined by the reviewer the inference made from these estimates did not change and the effect size changed only marginally.

Main text change

181-185

Planned pairwise contrasts revealed that self-reported pain was lower in the nature vs. urban [$b = -0.54$, $SE = 0.12$, $t_{(48)} = -4.46$, $p < 0.001$ one-tailed, 95% CI = [-0.782, -0.296], $d_{rm} = -0.53$] and indoor condition [$b = -0.48$, $SE = 0.11$, $t_{(48)} = -4.14$, $p < 0.001$ one-tailed, 95% CI = [-0.708, -0.245], $d_{rm} = -0.44$], with urban and indoor conditions not differing [$b = 0.06$, $SE = 0.12$, $t_{(48)} = 0.52$, $p = 0.60$, 95% CI = [-0.178, 0.303], $d_{rm} = 0.06$].

192-201

Specifically, planned pairwise contrasts revealed a significant difference in intensity ratings between nature vs. urban [$b = -0.25$, $SE = 0.12$, $t_{(48)} = -2.14$, $p = 0.018$ one-tailed, 95% CI = [-0.482, -0.015], $d_{rm} = -0.26$] and nature vs. indoor [$b = -0.29$, $SE = 0.11$, $t_{(48)} = -2.67$, $p = 0.005$ one-tailed, 95% CI = [-0.514, -0.073], $d_{rm} = -0.31$] but not for urban vs. indoor [$b = -0.05$, $SE = 0.12$, $t_{(48)} = -0.38$, $p = 0.71$, 95% CI = [-0.283, 0.193], $d_{rm} = -0.05$]. Similarly, the unpleasantness ratings showed a significant difference comparing nature vs. urban [$b = -0.83$, $SE = 0.16$, $t_{(48)} = -5.23$, $p < 0.001$ one-tailed, 95% CI = [-1.149, -0.511], $d_{rm} = -0.65$] and nature vs. indoor [$b = -0.66$, $SE = 0.15$, $t_{(48)} = -4.35$, $p < 0.001$ one-tailed, 95% CI = [-0.956, -0.355], $d_{rm} = -0.48$], but again not when comparing urban vs. indoor [$b = 0.17$, $SE = 0.15$, $t_{(48)} = 1.12$, $p = 0.27$, 95% CI = [-0.135, 0.475], $d_{rm} = 0.13$].

15. Reviewer: Did the authors control for the level of immersion in the different conditions? If participants more immersed in the nature, as compared to the other conditions this would be an important bias, or at least mechanistic explanation.

Response to reviewer:

We appreciate the reviewer's comment about potential effects of immersion. We have now examined differences in immersion ratings. In general, we find differences in perceived immersion between the three environments but using immersion as a covariate in our LMMs does not reveal a significant impact of immersion on pain outcomes and does not change the main or interaction effects of environment on our pain measures. While we prefer not to speculate extensively on their implications, due to the lack of preregistration for these effects, we now document the analyses and discuss them in the Supplementary Information (see lines 620-658) and reference them briefly in the main manuscript (see lines 294-297).

Main text change

294-297

We also performed further exploratory analyses on the association between self-reported and neural pain responses, as well as the role of immersion. Details on these analyses, the results, and their interpretation are documented in the Supplementary Results.

Supplementary Information text change:

620-658

Second, we examined whether the three environments differed in terms of immersion. Following previous research²⁵ we assessed subjective immersion using three items: Presence ("I felt present in the environment shown"), involvement ("I have forgotten the real environment around me"), and realism ("The environment shown felt very realistic"). Each item was rated on a scale from 1 ("not at all") to 5 ("very"). We calculated the average score of the three items (Cronbach's $\alpha = .81$) to estimate the overall immersion for each environment. To test for differences in immersion, we ran a LMM with the average immersion score as the dependent variable and environment as the predictor (nature as a reference, with random intercepts per participant). We found a significant effect of environment [$F_{(2,96)} = 16.62$, $p < 0.001$]. Post-hoc pairwise comparisons revealed that the nature environment was significantly more immersive compared to the urban [$b = 0.28$, $SE = 0.11$, $t_{(96)} = 2.54$, $p = 0.034$, 95% CI = [0.017, 0.541], $d_{rm} = .36$], or the indoor environments [$b = 0.63$, $SE = 0.11$, $t_{(96)} = 5.75$, $p < 0.001$, 95% CI = [0.371, 0.894], $d_{rm} = .80$]. Additionally, the urban environment was rated as significantly more immersive than the indoor environment [$b = 0.35$, $SE = 0.11$, $t_{(96)} = 3.22$, $p = 0.005$, 95% CI = [0.092, 0.616], $d_{rm} = .47$].

Third, to explore whether immersion levels explained differences in neural and self-reported responses to pain, we ran several additional LMMs. For these LMMs we used immersion (centered) as a covariate in all robust pain responses identified in our main analyses (see "Sensitivity Analysis" above). For immediate self-reported pain, we conducted a LMM with pain ratings as the dependent variable, to be predicted by the fixed effect of immersion, environment (nature as a reference), rating content (intensity as a reference), and the interaction of environment*rating content (with random intercepts and slopes for environment, rating content, and their interaction). For the signature responses, we performed a similar LMM with signature responses as the dependent variable, to be predicted by the effect of immersion, environment (nature as a reference), signature (NPS as a reference), and the environment*signature interaction (with random intercepts and slopes for environment and signature). Additionally, for the Thalamus, S2, and pINS, we ran LMMs using the ROI response as the dependent variable, to be predicted by the fixed effect of immersion, environment (nature as a reference), hemisphere (left hemisphere as a reference), and the environment*hemisphere interaction (with random intercepts and slopes for environment and hemisphere).

The results of the analyses revealed no significant main effect of immersion for the immediate ratings, signature, S2 and pINS models (all $p > .19$). For the Thalamus, immersion showed a trend-level result of $p = .067$. However, for all models the previously identified significant main and interaction effects of environment stayed significant. Thus, although the three environments showed differences in their absolute levels of self-reported immersion, we found no indication that immersion levels contributed significantly to the observed differences in self-reported and

neural responses to pain. Additionally, we found the same main and interaction effects of environment on pain after controlling for levels of immersion. These findings support the conclusion that immersion likely did not drive the observed effects.

16. Reviewer: Figure 1: the timing under the Video should be 3.5+/- 0.5.

Response to reviewer:

We have now corrected the values to 3.5 +/- 1.5 in the revised manuscript (see Figure 1).

Main text change:

861-875

Fig. 1 Stimuli and trial structure of the experiment. **a** Stimuli depicting a natural, an urban, and an indoor environment. A matching soundscape accompanied each visual stimulus. The three pain runs had a total duration of 9 min each, during which one environment was accompanied by 16 painful and 16 nonpainful shocks. All participants were exposed to all environments (in counterbalanced order). **b** Structure and timeline of an example trial. First, a cue indicating the intensity of the next shock (red = painful, yellow = not painful) was presented for 2,000 ms. Second, a variable interval of 3,500 ± 1,500 ms was shown. Third, a cue indicating the intensity of the shock was presented for 1,000 ms, accompanied by an electrical shock with a duration of 500 ms. Fourth, a variable interval of 3,500 ± 1,500 ms followed. Fifth, after each third trial, participants rated the shock's intensity and unpleasantness at 6,000 ms each. Sixth, each trial ended with an intertrial interval (ITI) presented for 2,000ms. The environmental stimulus was presented simultaneously except for the rating phase during each trial. Electrical painful and non-painful shocks were administered to the dorsum of the left hand with a separate electrode.

17. Reviewer: Discussion: The authors may want to consider discussing the role of prior experiences and learning mechanisms in the effect of nature on well-being, as well as the role of learned differences in social aspects in the different conditions.

Response to reviewer:

We appreciate the reviewer's thoughtful observation regarding the role of prior experiences and learning mechanisms, particularly in understanding the salutary effects of specific environments. In the Supplementary Information (see lines 568-591), we now discuss potential impacts of such experiences in conjunction with our findings, which show the absence of a moderating effect of nature connectedness. Although we did not directly assess prior experiences, nature connectedness may serve as a proxy for such experiences (Stehl et al., 2024, doi: <https://doi.org/10.1016/j.jenvp.2023.102225>). Given that we found no moderating role of nature connectedness, we argue that prior experiences were not fundamental to the effects observed in our study. However, we also stress that further research is necessary to test for these associations more directly.

Supplementary Information text change:

568-591

Thus, contrary to our preregistered hypothesis, our data did not reveal that nature connectedness significantly altered the impact of nature on self-reported and neural responses to pain. This contrasts with the literature suggesting that the beneficial effects of nature are moderated by nature connectedness¹⁸, a discrepancy that may arise from differences in outcome variables or the type of nature exposure investigated. Previous studies demonstrating this moderation effect focused on measures of mental health and well-being and examined the effect of real-world nature exposure. It is possible that the effects on mental health and well-being, which are less

tangible than immediate pain experiences in response to a specific stimulus, may be more pronounced. Furthermore, virtual nature may not provide the same increased benefits for individuals with higher levels of nature connectedness, as it could differ significantly from their typical experiences in nature. However, employing non-real, virtual nature rather than real-world stimuli also minimized potential confounding influences stemming from familiarity and prior experiences with specific settings. Notably, such prior experiences and learned associations may affect the salutary effects of nature. It has been suggested that positive past interactions can shape the restorative effects of nature through conditioning processes²⁰. When positive emotions are repeatedly linked with specific environments, cues from these environments may eventually elicit similar positive responses. Additionally, positive past experiences are related to high levels of nature connectedness which in turn are associated with greater perceived benefits from nature²¹. Since we found no moderating effect of nature connectedness – an outcome typically associated with positive prior experiences - on the pain outcomes, our current findings do not appear to support these suggestions. However, these interpretations are speculative, especially as nature connectedness is only an indirect measure of positive past experiences. Thus, future studies should investigate more directly the influence of prior experiences and learning mechanisms on the pain-relieving effects of nature.

18. Reviewer: The authors should temper their conclusions regarding the non-significant effects on the SIIPS1, as this outcome was not part of the original preregistration and it is debatable to draw conclusions from the lack of significance.

Response to reviewer:

In response to the reviewer's feedback, we have tempered the conclusion regarding the non-significant effects on the SIIPS1 (see lines 321-324 and 385-397). However, we would like to emphasize that the original manuscript had already contained a passage noting that the SIIPS1 was not preregistered, providing context for our originally planned analysis, which is reported in the Supplementary Information. We acknowledge that this may not have been explicit or visible enough, and hope that the text revisions have now enhanced clarity.

Main text change:

321-324

While this further supports the idea that nociception-related rather than cognitive-emotional aspects underpinned the subjective analgesic effects, it is important to note that the SIIPS1 analysis was not preregistered and should, therefore, be interpreted more cautiously.

385-397

However, it is important to approach our interpretation of the SIIPS1 response with caution. On the one hand, viewing the SIIPS1 as reflecting domain-general aspects is contingent upon the shortcomings of previous evidence validating its specificity to pain. Consequently, as suggested elsewhere³⁰, we propose that – unlike the NPS – the SIIPS1 may be influenced by cognitive and affective processes that are not specifically or exclusively associated with pain. On the other hand, we only preregistered the investigation regarding the NPS but not the SIIPS1 since we had originally planned to disentangle which pain components are predominantly affected using pooled ROIs activity. This decision was adopted later, but before looking at the data, because a direct comparison between signatures upon further reflection seemed more parsimonious and valid (see Supplementary Methods for further rationale). Thus, the selective effects on the NPS require further confirmation, as does the specific interpretation of the NPS versus the SIIPS1 in tracking nociception-related vs. domain-general responses engaged during the experience of pain.

19. Reviewer: In the discussion, the authors suggest that the positive effects of nature stimuli are due to the nature of the stimuli themselves, rather than the aversiveness of the urban control stimuli. However, I did not find data in the current report to support this claim. Was the aversiveness of the urban images measured? If so, could the authors provide additional data or clarification to support this assertion?

Response to reviewer:

We thank the reviewer for pointing out that this section was not formulated clearly enough. Our argument centers on the observation that prior studies have predominantly compared nature with aversive environments, leaving it unclear whether the observed differences were due to reduced pain in nature or heightened pain in the urban setting. Therefore, the inclusion of a neutral "baseline" condition against which we can compare the urban and nature conditions is key. Compared to this third "baseline" condition we observe that nature leads to lower pain rather than urban leading to greater pain (see also the updated information regarding the matching of the stimuli). Thus, by comparing nature with a relatively positive urban environment and an additional control condition, we can more confidently conclude that the difference is due to lower pain in the natural environment, rather than greater pain in the urban setting. This conclusion is reinforced by the consistent finding of nonsignificant differences between the urban and indoor environments: If the urban setting had indeed been aversive, we would have expected greater pain to be experienced than in the indoor "neutral" one. We have made this more explicit now in the discussion (see lines 333-347).

Main text change:

333-347

Importantly, unlike most past work, we used pre-tested and published stimuli of closely matched natural and urban settings (Materials and Methods) both of which have been rated comparably in terms of perceived beauty³⁴. Specifically, the urban stimuli contained many appealing and attractive elements from the nature scene, reducing the possibility that any differences would result from merely creating a spatially unmatched, noxious, and aesthetically displeasing urban setting^{13,14}. Nevertheless, we cannot rule out the possibility that the urban environment may have been perceived as generally aversive. Importantly, however, the self-reported and neural pain responses to the urban setting were broadly comparable to those in the more “neutral” indoor setting. This indicates that the predominant driver of differences in pain processing between the nature and urban settings was not the assumed higher aversiveness of the urban environment. Rather, it suggests that the observed pain differences between the urban and nature settings stem from the positive effects of the nature setting. Further support for this interpretation comes from the consistent patterns across both immediate and retrospective pain ratings, suggesting a similar pattern of differences across conditions.

Reviewer #2 (Remarks to the Author):

The manuscript presents a preregistered fMRI study (N = 49) investigating the behavioral and neural understanding of the analgesic effects of nature exposure on pain induced by electrical stimulation. The pre-registration of this study is commendable, as it strengthens its transparency and credibility. The authors employed well-controlled audiovisual stimuli to compare the effects of nature exposure with those of urban and indoor (neutral) virtual environments. They evaluated the impact of nature exposure on both neural and behavioral responses to noxious stimuli, finding that nature-related audiovisual stimuli (i.e., nature video and sound) reduced pain intensity and unpleasantness ratings compared to other control conditions, urban and neutral stimuli. Furthermore, analyses using fMRI pattern-based markers and ROIs revealed that these analgesic effects were associated with the changes in the main nociceptive processing system in the brain. Overall, this study provides evidence for the brain modulation effects of natural exposure for pain modulation. There are a few things that can improve the study, though.

Author response: We would like to sincerely thank the reviewer for the time and effort invested in improving our work and in the careful and attentive reading of our manuscript this entailed. We very much appreciate the reviewer’s thoughtful and constructive comments, particularly the suggestion for a more nuanced stance on some of our interpretations. Incorporating the suggestions has not only led to further corroboration of our findings, but also significantly contributed to enhance the overall quality of our manuscript.

Major comments:

1. Reviewer: The connection between the paper’s theoretical focus and the neuroimaging results could be improved. While the authors frame their research around the ART and SRT, suggesting that attentional or affective processes are key to understanding nature’s analgesic effects, it remains unclear how the fMRI markers (NPS and SIIPS1) map onto these cognitive processes.

Response to reviewer:

We appreciate the reviewer’s comment, which encouraged us to express our original intentions more clearly. In particular, we have made additional efforts to more directly connect these theoretical approaches from different fields in the discussion sections (see lines 428-442 and 456-463). It is, however, important to note that ART, SRT and pain theory have emerged from distinct disciplines, each with its own terminology and research foci. This divergence makes direct connections somewhat challenging. We now address this challenge explicitly and invite further research to align these perspectives in terms of theorizing and empirical corroboration. Within the context of our current findings, we can state that they align more closely with expectations set forth by ART, and we aim to make this connection clearer in the discussion section. We thank the reviewer for prompting us to do so with some more courage in our arguments.

Main text changes

428-442

Regarding nature’s potential to alleviate pain, the interpretation that reduced NPS activity is indicative of alterations in attention is particularly intriguing. In the field of environmental psychology, two prominent theories provide indirect frameworks for explaining nature’s analgesic effects on pain. On the one hand, stress recovery theory (SRT) posits that nature primarily influences affective responses⁹. In the context of pain research, SRT would imply that nature’s impact on pain should be explained by altered affect and reduced activity in higher-level neural processing linked to these affective changes (e.g., SIIPS1, PFC, or aMCC responses). As we did not observe such differences, our data offer limited support for such an interpretation. On the other hand, attention restoration theory (ART) suggests that natural stimuli can “restore” depleted attentional capacities⁹. The reasoning behind this argument is that nature possesses many features that are “softly fascinating” to humans and engage us in a distracting but not overly demanding manner. In the context of the experience of pain, ART would imply that features of the nature stimuli divert attention away from the painful sensation. In conjunction with findings from neuroscientific pain research, the observed reduction in nociception-related responses (e.g., the NPS, thalamus, S2, and pINS) substantiates this interpretation in favor of ART in two ways.

456-463

Notwithstanding these arguments, it is important to acknowledge that ART, SRT, and pain theories originate from distinct disciplines, each with unique terminologies and research foci. As a result, establishing direct connections between them remains somewhat imprecise at present and will require theoretical refinement and alignment in future work. We note that the attention regulation mechanism and the precise pattern of results were not specifically preregistered. The postulated interaction between attention- and nociception-related processes thus needs confirmation and extension by future research, which should focus on identifying how exactly attention-related brain areas act as regulators of the nociceptive inputs.

2. Reviewer: Relatedly, the authors interpret the NPS as nociception-related and SIIPS1 as domain-general cognitive-emotional. However, how do they ensure that this interpretation is correct? The original paper by Woo et al. (2017) does not clearly make this distinction. Rather, SIIPS1 is described as predicting pain above and beyond nociceptive input and mediating the pain-modulating effects of psychological factors such as expectations and perceived control. The interpretation of SIIPS1 as “domain-general cognitive-emotional” seems to require further justification or evidence beyond what is provided by the original paper.

Response to reviewer:

We thank the reviewer for highlighting the need for a more nuanced discussion regarding the NPS and SIIPS1. We acknowledge that the previous version of the manuscript may not have provided enough discussion on this topic. To address this, we have expanded our explanation and added new text elements to clarify this distinction in the introduction and discussion section of the main text (see line see lines 136-146 and 383-397).

In addition, we would also like to elaborate in more detail in our direct response to the reviewer here on why we maintain our original interpretation of the SIIPS1 as representing cognitive-emotional or domain-general aspects related to but unspecific to pain, while the NPS reflects nociceptive pain-specific processing. We also provide further arguments supporting the view that, compared to the NPS, the SIIPS1 may indeed track domain-general and cognitive-emotional processes, which are engaged during the experience of pain, rather than tracking pain-specific processes per se. Of note, we now better emphasize that we do not view the SIIPS1 as inherently domain-general. Rather we believe it reflects domain-general processes that are not specific to pain, but related to it, as more specifically noted now on several occasions in the main text (e.g., “domain-general brain responses to acute pain”, “domain-general processes while people experience acute pain”, “domain-general aspects of pain”, etc.).

First, (very) recent empirical studies which appeared after the original SIIPS1 paper and investigated different pain management strategies using the NPS and SIIPS1 align with our interpretation. Importantly, these studies were conducted and co-authored by the original developers of these signatures, further supporting our confidence in this distinction. For instance, Botvinik-Nezer and colleagues (2024, doi: 10.1038/s41467-024-50103-8) describe the NPS and SIIPS1 as representing “...nociceptive and affective pain neuromarkers...” respectively (p. 2). They also state that the NPS “tracks the intensity of nociceptive input ... and is largely specific to nociceptive pain...” (p. 2), that the SIIPS1 represents a “neuromarker for higher-level endogenous contributions to pain...” (p. 2), and that pain is “... more than nociceptive processing, and includes higher-level cognitive and affective processes...” (p. 6), with the latter being assessed by analyzing the SIIPS1 response. Similarly Wielgosz and colleagues (2022, doi: 10.1176/appi.ajp.21020145) interpret the NPS and SIIPS1 as reflecting “... direct, stimulus-related nociceptive activity and stimulus-independent elaborative cognition...” (p. 8) respectively. They also note that the SIIPS1 “...incorporates a range of cognitive and emotional modulatory circuits.” (p. 3). Thus, our current interpretation of these signatures aligns with the interpretation provided by the researchers who co-developed them.

Second, there is substantial evidence supporting the specificity of the NPS to nociceptive and lower-level pain processing. The NPS is highly responsive to noxious stimuli, accurately encoding their intensity, and shows minimal sensitivity to non-nociceptive aversive experiences. It also plays a limited role in higher-level pain modulations such as reappraisal, pain anticipation or placebo effects (see Reddan & Wager, 2018 for review, doi: 10.1007/s12264-017-0150-1). In contrast, the SIIPS1 was explicitly designed and trained to explain variance in pain after accounting for nociceptive processing. While it includes regions associated with nociceptive processing (e.g., pINS, S2), the weights of these regions in the SIIPS1 are not associated with the weights of these regions in the NPS (Woo et al., 2017, doi: 10.1038/ncomms14211). Moreover, the SIIPS1 incorporates a broad array of brain regions that are not predominantly linked to nociception, such as parts of the PFC, the cingulate cortex, caudate or nucleus accumbens. These areas are associated with rather overarching cognitive and affective functions, speaking for the interpretation that the SIIPS1 tracks the engagement of domain-general processes during the experience of pain. Furthermore, the SIIPS1 has been shown to mediate the effects of domain-general psychological interventions, such as expectancy manipulations, which influence pain through cognitive and affective pathways (Woo et al., 2017, doi: 10.1038/ncomms14211).

Third, current research provides more robust evidence for the NPS being specifically related to pain and less influenced by domain-general processes than for the SIIPS1. The NPS reliably responds to various forms of acute pain (thermal, visceral, mechanical, electrical) but does not respond to non-noxious aversive stimuli (see above). In contrast, to our knowledge, there is no existing evidence confirming whether the SIIPS1 shares this lack of response to non-noxious aversive stimuli. The SIIPS1 was also designed to capture a broad range of non-nociceptive aspects of pain and includes regions involved in diverse cognitive and affective processes. This broader scope makes the SIIPS1 inherently less specific to pain and more susceptible to influences from other non-noxious factors. Thus, the current lack of evidence supporting the pain specificity of the SIIPS1 further highlights that it represents domain-general aspects of the pain experience.

Nonetheless, notwithstanding these extensive findings and interpretation (and apologies for making the reviewer read such an extensive reply, but we wanted to make sure our arguments are built on solid grounds), we acknowledge that it may still be up for debate regarding their conclusiveness and exclusiveness. Accordingly, we have thus toned-down claims that could be perceived as overly strong or controversial, and added some more information on the processes the NPS and SIIPS1 intend to capture (see lines 136-146 and 383-397).

Main text changes

136-146

Importantly, the NPS has been shown to predict pain individually with high sensitivity and specificity, allowing the disambiguation from domain-general and non-specific processes such as negative emotion or cognitive appraisal, which also play a role in pain processing²⁷. In contrast, the SIIPS1 was explicitly developed to capture variance in pain after accounting for sensory processing. It primarily includes brain regions linked to cognitive and affective functions, mediates expectancy effects on pain, and may be shaped by broader cognitive and affective processes that are not exclusively tied to pain^{28,30}. Thus, it rather captures the engagement of domain-general processes while people experience acute pain. The aim of our study was to exploit these recent methodological developments and neuroscientific insights to better understand the neural processes and mechanisms by which nature exposure might lead to the reduction of painful experiences.

383-397

Therefore, it is noteworthy that this signature, and how it tracked the acute pain we exposed our participants to, was not significantly influenced by the nature vs. control stimuli. However, it is important to approach our interpretation of the SIIPS1 response with caution. On the one hand, viewing the SIIPS1 as reflecting domain-general aspects is contingent upon the shortcomings of previous evidence validating its specificity to pain. Consequently, as suggested elsewhere³⁰, we propose that – unlike the NPS – the SIIPS1 may be influenced by cognitive and affective processes that are not specifically or exclusively associated with pain. On the other hand, we only preregistered the investigation regarding the NPS but not the SIIPS1 since we had originally planned to disentangle which pain components are predominantly affected using pooled ROI activity. This decision was adopted later, but before looking at the data, because a direct comparison between signatures upon further reflection seemed more parsimonious and valid (see Supplementary Methods for further rationale). Thus, the selective effects on the NPS require further confirmation, as does the specific interpretation of the NPS versus the SIIPS1 in tracking nociception-related vs. domain-general responses engaged during the experience of pain.

3. Reviewer: In addition, the authors also dichotomize the brain regions' functions: pINS and S2 as early low-level nociception-related processing vs. aMCC and PFC as high-level emotional and motivational aspects. This dichotomy oversimplifies the complexity of these regions' roles in pain processing. For example, pINS and S2 are also included in SIIPS1. Similarly, areas like the PFC and aMCC are not exclusively linked to emotional and motivational processing but also play roles in sensory discrimination and attentional modulation. A more nuanced interpretation would reflect the integration of both sensory and affective processes within these regions, rather than a strict dichotomy.

Response to reviewer:

We appreciate the reviewer's important point regarding the complexity of pain processing and the need to emphasize a nuanced perspective on the functions of specific brain regions. We fully agree that pain processing is a complex phenomenon and that a strict dichotomy may not reflect this complexity. We have adopted a more nuanced interpretation in the introduction and discussion, have placed the terms 'lower-level' and 'higher-level' in quotation marks when introducing them in the text to highlight their heuristic nature, and added a more nuanced perspective as an additional point in our limitation section (see lines 107-124, 410-412 and 497-503).

In the context of ongoing debates regarding the precise roles and mapping of brain areas involved in cognitive, affective, and sensory processes, as well as the challenges of brain imaging in supporting such mappings, we acknowledge that caution is warranted when interpreting these activations. That said, the convergence observed across the neuroimaging analyses and subjective reports instills confidence that nature primarily influences lower-level nociceptive processing; we would thus like to share some more specific arguments with the reviewer in response to the specific points:

First, as detailed above, we see compelling evidence that the NPS is rather specifically tracking nociceptive and lower-level pain processing, which supports the argument that the mechanism by which nature affects pain is through alterations in nociceptive processing.

Second, the pain literature demonstrates varying degrees of specificity for nociceptive processing across different brain regions. That said, areas such as the thalamus, S2, and pINS are particularly well-supported in terms of their roles in sensory and nociceptive aspects of pain. The thalamus is a well-established relay station, transmitting nociceptive signals from the spinal cord to cortical areas (Apkarian et al., 2005, doi:

10.1016/j.ejpain.2004.11.001). Similarly, the pINS and S2 are strongly associated with sensory discriminative aspects of pain. The pINS, for instance, integrates sensory information from the body and serves as a crucial relay for nociceptive input, combining it with other somatosensory information (Mercer-Lindsay et al., 2021, doi: 10.1126/scitranslmed.abj7360). This integration is essential for processing the localization and intensity of a painful stimulus. Moreover, both the pINS and S2 have been shown to preferentially activate in response to noxious aversive as opposed to non-noxious aversive stimuli (such as sound), even when controlling for stimulus salience, highlighting their specificity in processing nociceptive information (Horing et al., 2019, doi:

10.1371/journal.pbio.3000205). However, we agree with the reviewer that these findings do not suggest that these areas are exclusively tracking or are specific to nociception - but, as we would argue, they indicate a strong preferential engagement during nociceptive processing.

Third, we also agree that areas such as the PFC or aMCC play a complex role in pain processing, but we are sure the reviewer will agree that their role predominantly relates to aspects that are 'farther away' from the initial, sensory aspects of pain processing. For instance, while there is an ongoing debate about the role of the ACC, there is converging evidence supporting its involvement in affective motivational aspects of pain processing (for reviews see: Apkarian et al., 2005, doi: 10.1016/j.ejpain.2004.11.001; Mercer-Lindsay et al., 2021, doi: 10.1126/scitranslmed.abj7360; Moraux et al., 2011, doi: 10.1016/j.neuroimage.2010.09.084; Tan & Kuner, 2021, doi: 10.1038/s41583-021-00468-2). Furthermore, the PFC seems to act as a hub that integrates various cognitive and affective sources of information with the incoming pain signals. It has been suggested to impact sensory discrimination indirectly via changes in cognitive (e.g., expectation, appraisal) and affective (e.g., mood, motivation) processes (Ong et al., 2019, doi: 10.1007/s12035-018-1130-9).

Again, we apologize for providing such an elaborate response, but we intend to make sure that our arguments and the evidence they are based on are clearly spelled out. We, thus, hope these arguments and their brief inclusion in the manuscript better explains our stance and we appreciate the constructive comment that prompted us to address them more explicitly and with added clarity in the revision.

Main text changes:

107-124

For example, while the posterior insula (pINS) and the secondary somatosensory cortex (S2) are **predominantly involved in early 'lower-level' nociception-related processing, 'higher-level' components incorporating sensory, emotional and motivational aspects** are associated with regions such as the anterior midcingulate (aMCC) and the prefrontal (PFC) cortex^{20,21}. **Distinguishing neural processes as pertaining to 'lower-level' vs. 'higher-level' is a useful yet necessarily imprecise heuristic (which is why we put them under quotation marks here).** Although individual brain regions rarely serve a singular role in pain processing, typically integrating both lower- and higher-level pain components²², evidence indicates that certain regions specialize in one aspect of pain over the other²³⁻²⁶. For instance, the pINS responds to noxious stimuli with minimal latency, is highly interconnected with sensorimotor cortices, and primarily encodes both the intensity and bodily location of a stimulus. In contrast, the anterior insula (aINS) exhibits a delayed response, is more strongly connected with prefrontal regions, and integrates information from the pINS to generate an emotional response to pain^{23,26}. **While acknowledging the complexity of pain processing and recognizing that the activation of brain areas aligns more with a gradient than a supposed dichotomy of underlying computations, evaluating brain responses during acute pain could yield more refined and less subjective assessments of the various processes underpinning the multifaceted quality of pain and help to disentangle if lower- or higher-level processes are impacted.**

410-412

While the engagement of any brain area during complex experiences, including pain, likely does not adhere to a singular function or a dichotomous functional organization, this remains an important finding.

497-503

Furthermore, dichotomizing brain regions as either sensory-discriminative ('lower-level') or affective-motivational ('higher-level') may oversimplify their roles, which are often multifaceted. Indeed, some authors challenge this separation, highlighting that evidence for distinct modulations of these components remains inconclusive²². Thus, while caution is warranted when interpreting regional and patterned brain activations, the convergence across neuroimaging analyses strengthens the argument that the nature stimuli primarily affected 'lower-level' components.

4. Reviewer: Please add figures for ROI analyses. At least the authors should show the location of ROIs to interpret their results more thoroughly.

Response to reviewer:

In line with the suggestion from the reviewer we now added two figures to the Supplementary Information (see lines 667-680 and 682-696) and reference them in the main text (see lines 247, 258, and 672) depicting the included ROIs (Supplementary Figure 2) and the planned pairwise comparisons separated by different pathways according to the framework by Bushell et al., 2013 (Supplementary Figure 3).

Main text change:

247

(Materials and Methods and Supplementary Figure 2)

258

(Supplementary Figure S3)

672

We created the following preregistered set of sphere-based ROIs (center [\pm x, y, z]; sphere size; **Supplementary Figure S2).**

667-680

Supplementary Fig. 2 Regions of interest for comparing neural pain responses. The selection of regions of interest (ROIs; center $[\pm x, y, z]$; sphere size) was guided by a theory-based approach, focusing on regions involved in three key neural circuits as outlined in the framework proposed by Bushnell et al.²⁶. The first circuit represents the ascending pathway, which includes the left/right primary somatosensory cortex (S1; $[\pm 39 -30 51]$, 10mm) and the left/right thalamus ($[\pm 12, -18, 3]$; 6mm). The second circuit is linked to attentional modulations of pain. It involves the left/right superior parietal lobe (SPL; $[\pm 18, -50, 70]$; 10mm), the left/right secondary somatosensory cortex (S2; $[\pm 39, -15, 18]$; 10mm), the left/right posterior insula (pINS; $[\pm 44, -15, 4]$; 10mm), and the left/right amygdala ($[\pm 20, -12, -10]$; 10mm). The third circuit is associated with emotional modulation of pain, encompassing the left/right anterior insula (aINS; $[\pm 33, 18, 6]$; 10mm), the anterior midcingulate cortex (aMCC; $[-2, 23, 40]$, 10mm), the medial prefrontal cortex (mPFC; $[7, 44, 19]$; 10mm), and the periaqueductal gray (PAG; $[0, -32, -10]$; 6mm).

681-696

Supplementary Fig. 3 Activity differences in regions of interest across environments. Regions of interest (ROI) are organized in three key neural circuits as outlined in the framework proposed by Bushnell et al.²⁶. **a** Schematic representation of neural circuits and associated ROIs. Grey, blue, and yellow ROIs belong to the ascending, descending (emotion), and descending (attention) pain pathways respectively (adapted with permission from Bushnell et al.²⁶). **b** Bar-plot of mean contrast estimates and standard errors within the primary somatosensory cortex (S1) and the Thalamus per environment. **c** Bar-plot of mean contrast estimates and standard errors within the anterior insula (aINS), anterior midcingulate cortex (aMCC), medial prefrontal cortex (mPFC), and periaqueductal grey (PAG) per environment. **d** Bar-plot of mean contrast estimates and standard errors within the amygdala, posterior insula (pINS), secondary somatosensory cortex (S2) and superior parietal lobe (SPL). All means represent estimated marginal means (arbitrary units) derived from the linear mixed models. † < .1, * < .05, ** < .01, mark trend-level and significant pairwise comparisons; ‡ marks trend-level pairwise comparisons in absence of a significant main effect of environment in the linear mixed model. S.E.M. = standard error of the mean, a.u. = arbitrary units.

5. Reviewer: In the context of Stress Recovery Theory (SRT), positive affect is a key component, and thus the preregistered plan includes analyses on positive affect (e.g., using PANAS), but nothing related to positive affect is presented in the current manuscript. Given that positive affect may play an important role in the observed effects, particularly in supporting the SRT framework, it would be beneficial to include these results to provide a more comprehensive analysis.

Response to reviewer:

We thank the reviewer for this thoughtful comment, which we had already captured in our pre-registration. Due to space constraints, however, we had opted to not report these analyses. Based upon the reviewer's comment, we have reconsidered this decision and now include the additional preregistered analyses on positive and negative affect, alongside all other preregistered hypotheses, in the Supplementary Information. The PANAS ratings do show significant differences across all environments in the expected direction (e.g., lowest negative and highest positive affect in nature). However, this is not consistently related to the changes in subjective and neural responses to pain we observe (e.g., although urban and indoor environments differ in positive and negative affect, they do not differ regarding pain responses; see lines 287-294 in main text and 170-177 and 456-496 in Supplementary Information). We appreciate that these findings may have implications for SRT especially but do not really have space to elucidate on that in detail in the main text and hope that what we have now added is sufficient to accurately reflect the results we collected alongside some considerations of their implications.

Main text change:
287-294

Note that in addition to these analyses addressing our hypotheses related to pain outcomes, we had preregistered three additional hypotheses. In brief, we observed (1) that environments significantly differed regarding positive and negative affect (with nature showing higher positive and lower negative affect ratings when compared to urban or indoor settings), (2) that pulse rate was lower in the nature than in the urban setting, and (3) that nature connectedness, contrary to our prediction, did not moderate the main findings (i.e., participants who felt more psychologically close to nature did not show greater benefits compared to those who felt less connected).

Supplementary Information text change:
170-177

Lastly, three additional hypotheses had been part of our preregistration. These hypotheses focused on the impact of the different environments on positive and negative affect, pulse rate, and the potential moderating role of individual differences in nature-connectedness on the effects of nature. The results of these analyses and their interpretations are documented in the Supplementary Results section titled "Additional preregistered analyses". Notably, we observed significant differences in positive and negative affect between the environments, a reduction in pulse rate when comparing the nature and urban environments, and no evidence of moderation effects related to nature connectedness.

456-496

The effect of environments on positive and negative affect

Based on prior literature¹⁵ we preregistered that nature exposure would lead to heightened positive and reduced negative affect compared to urban or indoor environments. Additionally, we expected these effects to be more pronounced for indicators of positive than for negative affect.

To address these hypotheses, participants completed the Positive and Negative Affect Schedule (PANAS) after each run in the experiment¹⁶. The PANAS measures subjective levels of positive and negative affect using 10 items each. Each item represents a specific feeling, and participants indicate on a scale from 1 ("very slightly or not at all") to 5 ("extremely") how much the respective feeling applies to them at that moment. The two subscales of the PANAS – positive and negative affect – were calculated by averaging the items corresponding to each type of affect valence. To assess whether positive affect (PA) or negative affect (NA) differed across the three environments, we compared the PANAS ratings after each pain run, which included the presentation of painful and non-painful shocks alongside the environment. We calculated a LMM using affect (PA and NA) as the dependent variable, to be predicted by the fixed effect of environment (nature as a reference), valence (NA as a

reference), and their interaction (with random slopes and intercepts for environment, valence, and their interaction by participant).

We observed a significant main effect of environment [$F_{(2,111.58)} = 9.09, p < 0.001$], valence [$F_{(2,48.00)} = 114.36, p < 0.001$], and their interaction [$F_{(2,49.02)} = 16.63, p < 0.001$]. In line with our predictions, planned pairwise comparisons revealed significantly lower negative affect in nature compared to urban [$b = -0.33, SE = 0.06, t_{(48)} = -5.18, p < 0.001$ one-tailed, 95% CI = [-0.463, -0.204], $d_{rm} = -0.76$] or nature compared to indoor environments [$b = -0.21, SE = 0.06, t_{(48)} = -3.60, p < 0.001$ one-tailed, 95% CI = [-0.322, -0.092], $d_{rm} = -0.58$]. Additionally, positive affect was significantly higher in nature compared to urban [$b = 0.14, SE = 0.07, t_{(48)} = 1.98, p = 0.026$ one-tailed, 95% CI = [-0.002, 0.276], $d_{rm} = 0.21$] and nature compared to indoor environments [$b = 0.34, SE = 0.07, t_{(48)} = 4.78, p < 0.001$ one-tailed, 95% CI = [0.195, 0.478], $d_{rm} = 0.55$]. We also found significantly higher positive affect [$b = 0.20, SE = 0.07, t_{(48)} = 2.80, p = 0.007, 95\% \text{ CI} = [0.056, 0.343], d_{rm} = 0.33$], and higher negative affect [$b = 0.13, SE = 0.06, t_{(48)} = 2.02, p = 0.049, 95\% \text{ CI} = [0.001, 0.253], d_{rm} = 0.28$] for urban compared to indoor environments.

Thus, in line with our prediction, receiving painful stimulation while being exposed to nature was associated with higher positive and lower negative affect compared to exposure to urban or indoor environments. Contrary to our prediction, the effect was more pronounced for negative (moderate to high effect size), than for positive affect (low to moderate effect size). Since our study, unlike most previous nature-based research, investigated affective processing in an aversive context (i.e., painful electrical stimulation), we speculate that this may have resulted in greater variability in negative affective responses, allowing for more pronounced differences across environments. Furthermore, we also observed differences when comparing urban and indoor environments, although these effects were generally smaller and less consistent in terms of their direction. Although the observed differences in self-reported affect are consistent with previous findings and appear to support stress recovery theory (SRT), the changes in neural pain outcomes do not suggest that the reduction in pain was driven by these affective changes (see discussion in main text).

6. Reviewer: Relatedly, although the main findings focus on the differential effects between two fMRI pattern-based markers, only one marker (NPS) was preregistered. Furthermore, the relevant hypotheses in the preregistered plan were marked as 'non-directional,' indicating a significant portion of the study is still exploratory. This reduces the strength of the preregistration's contribution. For greater transparency, it would be helpful for the authors to clearly specify which analyses adhered to the preregistered plan, which were exploratory, and which preregistered analyses were omitted in the current study (e.g., positive affect and physiological responses, etc.).

Response to reviewer:

We appreciate the reviewer's comment regarding the preregistration and the need for more clarity on deviations from it. We have now included all hypotheses and their related findings in the Supplementary Information, with brief references provided in the main text (see lines 287-294). Additionally, we have made further clarifications to emphasize these points in the main text and outline all deviations from preregistration in detail in the Supplementary Information (see our section "Preregistration and deviations" in the Supplementary Information lines 63-66 and 133-179). Lastly, we highlighted the exploratory or confirmatory nature of the hypotheses in the main text on several occasions (e.g. "Exploratory analyses of these ratings revealed...", "The exploratory findings of the retrospective ratings...", "Complementary exploratory whole-brain analyses...", "further exploratory analyses on the association between self-reported and neural pain responses, as well as the role of immersion...", etc.)

Main text change:
287-294

Note that in addition to these analyses addressing our hypotheses related to pain outcomes, we had preregistered three additional hypotheses. In brief, we observed (1) that environments significantly differed regarding positive and negative affect (with nature showing higher positive and lower negative affect ratings when compared to urban or indoor settings), (2) that pulse rate was lower in the nature than in the urban setting, and (3) that nature connectedness, contrary to our prediction, did not moderate the main findings (i.e., participants who felt more psychologically close to nature did not show greater benefits compared to those who felt less connected).

Supplementary Information text change:
63-66

Preregistration and deviations.

Generally, we strictly adhered to our preregistration, registered on 12 May 2022 on OSF (osf.io/t8dqu), transparently report deviations, and label additionally performed and exploratory analyses as such.

133-179

Furthermore, to ensure the robustness of our MRI analyses, we conducted several sensitivity analyses across different analytical approaches. Specifically, we reanalyzed the data by: (1) excluding participants identified as statistical outliers, (2) applying a motion threshold of 2 mm, (3) using a more parsimonious first-level model, and (4) employing alternative masks for our ROIs. Detailed descriptions of the results are provided in the Supplementary Results section "Sensitivity analysis". Notably, despite these variations in analytical approaches, results for the NPS and key ROIs, including the thalamus, S2, and pINS remained consistent. However, greater

variability in the outcomes was observed for the SPL and aINS, suggesting that findings in these regions should be interpreted with caution.

Regarding points (1) and (2), the primary analyses presented in the main manuscript do not exclude statistical outliers or participants exceeding motion thresholds. Although we initially planned to exclude these cases, we reconsidered this approach (before performing any analyses or looking at any data) because outliers may capture meaningful individual variability and exclusions could compromise ecological validity. Moreover, residual movement was modeled as a nuisance regressor, and visual inspection of the data (orthogonal to any hypotheses, pain>no-pain) revealed that participants exceeding the 2mm movement criterion showed no major signs of movement artifacts. These considerations are particularly pertinent when analyzing multivariate patterns such as the NPS and SIIPS1. Upon reviewing the distribution of these pattern responses in prior research^{12,13}, we concluded that deviating values represent valid variability within our sample. Consequently, we opted to retain these data in our analysis. However, for full transparency and completeness of reporting all analyses, we report the results of analyses excluding these participants in the section “Sensitivity analyses” below.

Regarding point (3), we initially employed a parsimonious first-level model to reduce the risk of overfitting. This model included eight regressors: delivery of painful shocks, delivery of non-painful shocks, and six motion regressors. This approach was also outlined and reported in the preprint version of this article. Following reviewer feedback and in alignment with standard modeling approaches used in similar research^{12,14}, we revised the first-level model to incorporate three additional regressors. This revised more comprehensive model includes the following 11 regressors: anticipation of painful shocks, anticipation of non-painful shocks, delivery of painful shocks, delivery of non-painful shocks, ratings, and six motion regressors. This revised approach is reported in the main article, while the initial parsimonious model is detailed in the section “Sensitivity analyses” below.

Regarding point (4), the main article reports results based on ROIs that include only voxels showing a significant response to painful versus non-painful stimuli in the pain>no-pain contrast across all environments. Following a reviewer’s suggestion, we reanalyzed the data using the full ROI masks that encompassed all voxels irrespective of their sensitivity to pain. These additional analyses are detailed in the section “Sensitivity analyses” below. Lastly, three additional hypotheses had been part of our preregistration. These hypotheses focused on the impact of the different environments on positive and negative affect, pulse rate, and the potential moderating role of individual differences in nature-connectedness on the effects of nature. The results of these analyses and their interpretations are documented in the Supplementary Results section titled “Additional preregistered analyses”. Notably, we observed significant differences in positive and negative affect between the environments, a reduction in pulse rate when comparing the nature and urban environments, and no evidence of moderation effects related to nature connectedness. Furthermore, prompted by reviewer feedback, we also report non-preregistered exploratory analyses in the results section titled “Additional non-preregistered analyses”.

7. Reviewer: The connection between the main findings of the current study (e.g., NPS and SIIPS results) and some behavioral findings, such as distraction and tolerance, is not clearly established or explained.

Response to reviewer:

We thank the reviewer for the valuable feedback. In response, we have included correlation analyses between immediate self-reported pain and neural responses to pain, which we briefly discuss in the main text (see lines 294-297) and the Supplementary Information (see lines 597-619). Additionally, we briefly discuss how the ROI results and the retrospective pain ratings converge to support the claims implied by ART in the discussion (see lines 450-454). Importantly, the distraction and tolerance measures were not intended to support mechanistic explanations, as their retrospective nature introduces additional noise and memory bias, potentially leading participants to report in a socially desirable or compliant manner (as further evidenced by the substantially larger effects sizes of retrospective when compared to the immediate pain ratings). Thus, our primary focus was on the preregistered immediate intensity and unpleasantness ratings, which aimed to replicate and extend previous studies to establish whether nature indeed has a decreasing effect on pain.

Main text changes:

294-297

We also performed further exploratory analyses on the association between self-reported and neural pain responses, as well as the role of immersion. Details on these analyses, the results, and their interpretation are documented in the Supplementary Results.

450-454

Second, asking participants if exposure to the respective environment helped to distract themselves from pain revealed effect sizes in the medium to high range when comparing nature to urban ($d_m = .66$) or indoor settings ($d_m = 1.04$) while comparing urban and indoor stimuli ($d_m = 0.34$) showed only a small effect (Supplementary Results).

Supplementary Information text changes:

597-619

First, we conducted correlation analyses to examine the relationship between self-reported and neural responses to pain. Specifically, we aggregated immediate pain ratings – separated by intensity and unpleasantness – and neural responses for the NPS, SIIPS1, Thalamus, S2, and pINS, separately across environments. Our analyses focused on neural responses that demonstrated robust, significant differences between environments as reported

in the “Sensitivity Analyses” section. The correlation analyses revealed moderate positive associations between neural responses and self-reported pain intensity with correlations of $r = .32$ ($p = .024$) for the NPS and $r = .40$ ($p = .009$) for the SIIPS1 response. Similarly, these responses were moderately associated with unpleasantness ratings ($r = .49$, $p = .001$; and $r = .50$, $p = .001$; respectively). All associations between signature responses and ratings were significant after multiple comparison corrections across the four correlations (Bonferroni-Holm). We also observed weak positive, but non-significant correlations between the thalamus ($r = .09$, $p = 1$), S2 ($r = .20$, $p = .51$), and pINS ($r = .05$, $p = 1$) responses and pain intensity. Notably, these regions showed slightly stronger correlations with unpleasantness ratings ($r = .28$, $p = .25$; $r = .33$, $p = .12$; and $r = .24$, $p = .36$; respectively). Applying multiple comparison correction across the six correlations (Bonferroni-Holm) rendered the correlations between ROI responses and unpleasantness ratings as not significant anymore. Thus, higher self-reported pain was significantly associated with higher neural responses in the signature but not the ROI data. The magnitude of these associations is consistent with prior studies, which report weak to moderate associations between self-reported pain and neural responses to pain^{13,22}. These findings highlight the value of integrating both neural and self-reported pain measures, as they offer complementary insights into pain processing. While self-reported pain reflects the subjective experience, neural data can reveal the underlying brain mechanisms involved in pain perception^{23,24}.

8. Reviewer: Furthermore, to elucidate the detailed neural mechanisms of the analgesic effects induced by nature exposure, it would be beneficial to analyze other events (e.g., cue or video) within the experiment. This could help clarify whether the observed effects are stimulus-specific or persist throughout the trials.

Response to reviewer:

This suggestion from the reviewer is much appreciated as it aligns well with our plans for subsequent analyses. However, since these analyses were not included in our preregistration and thus are somewhat outside the main aims of our study, they need to be addressed from a more purely exploratory perspective. We are in the process to pre-registering an analysis plan and examining preliminary results at this stage would undermine our goal of adhering to a rigorous analysis framework. For this reason, we prefer to abstain from reviewing any results at this time. Furthermore, these analyses may also extend beyond the scope of the current manuscript, in particular when considering that space is limited by the journal and that we are already very close to the word limit. For these reasons, we have opted to not pursue such analyses for the current manuscript. Should the reviewer and editor deem it essential to include this information in the current manuscript, we are open to reconsider our decision.

Minor comments/questions:

9. Reviewer: I am not sure if “hypoalgesia” is the correct term for the pain-modulatory effects of nature exposure. Hypoalgesia is often used in clinical or pharmacological contexts. An alternative term could be “analgesic effects,” which is more generally used to describe the reduction in pain intensity or unpleasantness without necessarily implying a clinical or pharmacological mechanism.

Response to reviewer:

We appreciate the advice that the term hypoalgesia might have a clinical or pharmacological connotation. Considering this we adopted the term to “analgesic effects” as suggested by the reviewer. This has been changed on all occasions in the main text, including a slight amendment of the manuscript title, and Supplementary Information.

10. Reviewer: While Fig. 1B effectively outlines the experimental paradigms, it appears that the graphical illustration does not fully capture specific elements of the figure legend, e.g., “after each third trial, participants rated the shock’s intensity and unpleasantness at 6,000 ms each”. Please consider revising the figure to better align with the described procedures.

Response to reviewer:

We appreciate the reviewer for highlighting that the figure did not depict the experimental design clear enough. In response, we have revised the figure to ensure that it now unambiguously represents the design.

Main text change:

861-875

Fig. 1 Stimuli and trial structure of the experiment. **a** Stimuli depicting a natural, an urban, and an indoor environment. A matching soundscape accompanied each visual stimulus. The three pain runs had a total duration of 9 min each, during which one environment was accompanied by 16 painful and 16 nonpainful shocks. All participants were exposed to all environments (in counterbalanced order). **b** Structure and timeline of an example trial. First, a cue indicating the intensity of the next shock (red = painful, yellow = not painful) was presented for 2,000 ms. Second, a variable interval of $3,500 \pm 1,500$ ms was shown. Third, a cue indicating the intensity of the shock was presented for 1,000 ms, accompanied by an electrical shock with a duration of 500 ms. Fourth, a variable interval of $3,500 \pm 1,500$ ms followed. Fifth, after each third trial, participants rated the shock's intensity and unpleasantness at 6,000 ms each. Sixth, each trial ended with an intertrial interval (ITI) presented for 2,000ms. The environmental stimulus was presented simultaneously except for the rating phase during each trial. Electrical painful and non-painful shocks were administered to the dorsum of the left hand with a separate electrode.

11. Reviewer: In the results section, the interaction between environment and rating type for pain ratings was reported with degrees of freedom that include decimal points [$F(2,81.14) = 9.19, p < 0.001$], while similar analysis (interaction between environment and signature) with signature responses did not ($[F(2,96) = 6.04, p = 0.003]$). If these discrepancies result from different statistical assumptions (e.g., equal variance) applied across the linear mixed model (LMM) analyses, it would be helpful to clarify this in the methods or results sections.

Response to reviewer:

We thank the reviewer for the very attentive and accurate feedback. Indeed, almost all main effects of our LMMs have degrees of freedom with decimals. The decimal places are to be expected when using Satterthwaite's approximation, as we did for our models, and are not unusual. In many of our analyses the decimal places were close to or almost exactly zero, which is why they were rounded up/down. In the adapted manuscript we now report all degrees of freedom to exactly the second decimal place to avoid potential confusion. It is important to note, that these discrepancies do not result from different statistical assumptions but are due to rounding.

12. Reviewer: The manuscript currently lacks detailed information on the pain calibration task (e.g., the number of stimuli and the range). Please provide a description of the pain calibration task in the Method sections.

Response to reviewer:

In accordance with the suggestions from the reviewer, we have now included a brief description of the pain calibration procedure in the Methods section (see lines 583-589). This addition enhances clarity and completeness in the section.

Main text change:

583-589

The calibration consisted of three phases, separated by breaks of approximately 3 min, and was conducted inside the scanner. Participants received an initial low-intensity shock (0.05 mA) in the first two phases, followed by progressively stronger shocks rated on a scale from 0 to 8. Each phase concluded when a shock was rated as 8, resulting in a variable number of shocks for each participant. In the third phase, shocks reflecting the average intensities of 1 and 6 from the first two phases were pseudorandomly administered and rated by the participants to confirm the stability of their perceived intensity.

13. Reviewer: Regarding the first-level analysis, could you clarify whether only shock events were included in the design matrix, or if other event-related regressors (e.g., cue presentations and ratings) were also included in the design matrix? Additionally, please specify whether temporal filtering was applied or not.

Response to reviewer:

We thank the reviewer for highlighting that the details of our first-level analysis need clarification. Our original approach prioritized model parsimony, which is why we chose not to include additional event-related regressors beyond the delivery of painful and non-painful shocks to the design matrix, to avoid 'overfitting' the data. While the reviewer did not expressly criticize this approach, the comment prompted us to reconsider our scientific rationale. Consequently, we decided to assess the robustness of our findings by re-running all analyses using a first-level model that also modeled cue (for painful and non-painful stimuli separately) and rating as additional regressors. In brief, this analysis yielded results that were virtually identical to those resulting from the use of the more parsimonious model, providing converging evidence in the sense of a multiverse analysis (Botvinik-Nezer et al., 2020, doi: 10.1038/s41586-020-2314-9). In more detail, we first assessed whether the more comprehensive model led to a more precise and robust identification of the areas involved in pain processing, using the contrast Pain > No Pain (pooled across conditions, i.e. using a contrast that is orthogonal to and unrelated to our main environment-based hypotheses and thus would not introduce potential bias or circularity re: condition effects). As this was indeed the case (see Supplementary Information 230-265), we then continued to perform our pre-registered hypotheses testing condition effects with the more comprehensive first-level model. These analyses confirmed the same significant interaction effect between environment*signature for the signature data, as well as the same main effects of environment in the thalamus, S2 and, pINS. Additionally, planned pairwise comparisons revealed the same significant differences as in the original analyses. Apart from the excellent convergence and virtual identity of findings, the following arguments speak for using the more comprehensive model as the main analysis approach. In the more parsimonious model, the implicit baseline is a mix of non-modeled events; thus, including these events as regressors, the baseline becomes more readily interpretable. Moreover, upon re-examining the literature employing comparable study designs, we found this approach to be more prevalent (e.g. Botvinik-Netzer et al., 2024, doi: 10.1038/s41467-024-50103-8 ; Wielgosz et al., 2022, doi: 10.1176/appi.ajp.21020145). Nevertheless, we prefer to report both analyses and their findings, with the updated and original analysis included in the main text (see lines 238-240, 274-277, and 644-646) and Supplementary Information (see lines 133-141, 155-164, and 401-424), respectively. This ensures a fully transparent documentation of all analyses performed, communicates the adapted rationale for using the more comprehensive first-level model as the primary one in the end, and by way of the "multiverse" approach with full converging evidence strengthens our conclusions. Additionally, we also included the temporal filtering specification in the main text (see lines 650-651).

Main text changes

238-240

Importantly, these effects remained largely consistent after excluding statistical outliers, participants exceeding motion thresholds, and when applying an alternative first-level model specification for the MRI data (Supplementary Results).

274-277

Importantly, the effects in the thalamus, S2, and pINS remained largely consistent after excluding statistical outliers, participants exceeding motion thresholds, when applying an alternative first-level model specification for the MRI data, and when using alternative ROI masks (Supplementary Results).

644-646

A design matrix was specified with the following five experimental regressors per environment (i.e., run): anticipation of painful shocks, anticipation of non-painful shocks, delivery of painful shocks, delivery of non-painful shocks, and rating.

650-651

Furthermore, we applied SPM12's standard temporal filter methods, including the use of a high-pass filter with a default cut-off of 128 s to remove low-frequency noise.

Supplementary Information text changes

133-141

Furthermore, to ensure the robustness of our MRI analyses, we conducted several sensitivity analyses across different analytical approaches. Specifically, we reanalyzed the data by: (1) excluding participants identified as statistical outliers, (2) applying a motion threshold of 2 mm, (3) using a more parsimonious first-level model, and (4) employing alternative masks for our ROIs. Detailed descriptions of the results are provided in the Supplementary Results section "Sensitivity analysis". Notably, despite these variations in analytical approaches, results for the NPS and key ROIs, including the thalamus, S2, and pINS remained consistent. However, greater variability in the outcomes was observed for the SPL and aINS, suggesting that findings in these regions should be interpreted with caution.

155-164

Regarding point (3), we initially employed a parsimonious first-level model to reduce the risk of overfitting. This model included eight regressors: delivery of painful shocks, delivery of non-painful shocks, and six motion regressors. This approach was also outlined and reported in the preprint version of this article. Following reviewer feedback and in alignment with standard modeling approaches used in similar research^{12,14}, we revised the first-level model to incorporate three additional regressors. This revised more comprehensive model includes the

following 11 regressors: anticipation of painful shocks, anticipation of non-painful shocks, delivery of painful shocks, delivery of non-painful shocks, ratings, and six motion regressors. This revised approach is reported in the main article, while the initial parsimonious model is detailed in the section “Sensitivity analyses” below.

401-424

Parsimonious first-level model

We repeated all analyses using an additional first-level model for the MRI data. Our original approach prioritized a parsimonious first-level model, which included 8 regressors for each environment: one regressor for delivery of painful shocks, one regressor for delivery of non-painful shocks, and six nuisance regressors accounting for motion. This approach was initially presented in the preprint of this article. After receiving feedback from the reviewers, re-examining the literature on similar research questions and designs^{12,14}, and principled arguments regarding the interpretability of the used implicit baseline, we reconsidered this approach and implemented an additional, more comprehensive first-level model. This model included 11 regressors for each environment: one regressor for the anticipation of painful shocks, one for the anticipation of non-painful shocks, one for delivery of painful shocks, one for delivery of non-painful shocks, one for the rating phase, and six nuisance regressors accounting for motion. The findings using this more comprehensive model, following validation checks orthogonal to the main hypotheses (see main text), are now reported in the main text. For transparency and completeness, we also present the results of the more parsimonious first-level model here.

Using the more parsimonious model yielded the same significant interaction between environment*signature [$F_{(2,96.00)} = 6.04, p = 0.003$] for the signature data. Additionally, we observed the same significant main effects of environment in the Thalamus [$F_{(2,47.99)} = 5.53, p = 0.006$], S2 [$F_{(2,48.00)} = 5.16, p = 0.009$], and pINS [$F_{(2,48.00)} = 9.28, p = 0.0003$]. However, the previously significant and trend-level effects in the SPL [$F_{(2,48.00)} = 2.08, p = 0.136$] and aINS [$F_{(2,48.00)} = 2.39, p = 0.101$] were no longer observed.

Inspection of the planned pairwise comparisons for the significant main and interaction effects revealed the same overall results. There was no change in significance regarding the planned pairwise comparisons in the NPS, Thalamus, S2, and pINS.

14. Reviewer: Given the potential differences in visual and acoustic features between indoor, nature, and urban stimuli (especially between indoor and other stimuli), were any nuisance regressors included to account for these variations? Clarification on this point would help with the interpretation of the results.

Response to reviewer:

We appreciate the reviewer’s comment regarding the inclusion of additional nuisance regressors. Our approach prioritized ecological over internal validity by selecting diverse visual content and soundscapes that represent examples of real-world environments. While we acknowledge that differences in these features may influence our results, we did not include nuisance regressors to account for them. The rationale is that regressing out these features could diminish the sensitivity as well as the external validity of our findings, as real-life urban vs. nature environments seldom exhibit identical visual and acoustic characteristics. To more explicitly address this aspect, we now emphasize in the discussion (see lines 472-488) that the current study serves as an initial step in determining whether differences exist between environments in general, providing a steppingstone for future studies exploring the specific features and factors driving these effects.

Main text changes:

472-488

Second, more granularity is required to thoroughly assess which specific elements of nature are relevant in driving the observed analgesic effects. The literature on the benefits of nature suggests that certain perceptual features make natural settings particularly fascinating^{9,13}. These features might exhibit a notably engaging effect, thus leading to a stronger diversion from pain. Complex cognitive and emotional reactions, such as feelings of awe and nostalgia, towards these features might be essential⁴², but which particular feature is relevant remains unclear. However, our exploratory whole-brain analysis indicated that, despite being matched in loudness, the urban environment resulted in distinct processing in auditory cortex, compared to nature. This finding provides preliminary evidence that it may not only be the visual quality of the nature stimuli that makes them effective, but that variations in soundscapes may have downstream effects on pain processing as well. Notably, although differences in auditory processing may relate to alterations in pain, they are unlikely to be the primary driver of the observed effects. This is evidenced by the lack of differences in auditory processing between the nature and indoor environments, despite comparable magnitudes of change in pain outcomes across both the urban vs. nature and indoor vs. nature comparison. Thus, further work is needed to explore which specific sensory elements and their combination make natural environments particularly effective in alleviating pain.

15. Reviewer: According to the legend of Figs. 2A and 2B, it is addressed that each dot in the violin plots represents single values. Does the dot represent immediate self-report ratings for each trial? Please clarify.

Response to reviewer:

Yes, each dot in Figures 2A and 2B represents a single value, specifically the immediate self-report ratings provided by each participant for each trial (i.e., 6 self-reported ratings of intensity or unpleasantness for each participant in each environment). Should there be any questions regarding the single datapoint with a value of 4.2 for intensity in the indoor condition, we would like to clarify that our original analysis included an imputed value of

one missing data point. We now excluded this imputed value and treated it as missing, which led to almost identical results across analyses (see below).

Fixed Effect	With imputation	Without imputation
Environment	$[F_{(2,48.14)} = 12.48, p < 0.001]$	$[F_{(2,48.14)} = 12.49, p < 0.001]$
Rating content	$[F_{(1,48.00)} = 17.51, p < 0.001]$	$[F_{(1,48.00)} = 17.52, p < 0.001]$
Environment*Content	$[F_{(2,81.14)} = 9.19, p < 0.001]$	$[F_{(2,81.11)} = 9.19, p < 0.001]$

16. Reviewer: In Figs. 2D and 2F, the standardized betas are displayed rather than the signature response values. Could you provide a rationale for this choice?

Response to reviewer:

We appreciate the reviewer for bringing this error to our attention. The values displayed in Figures 2D and 2F are standardized signature responses, not betas. We have revised the wording in the manuscript to accurately reflect this (see lines 878-889).

Main text changes:

Fig. 2 Behavioral and pain signature responses across environments. Violin plots depicting **a** intensity and **b** unpleasantness ratings of painful shocks and the overall lower-level nociceptive **d** and higher-level cognitive-emotional **f** neural response to pain as indicated by the neurologic pain signature (NPS, **c**) and the stimulus intensity independent pain signature-1 (SIIPS1, **e**) for each environment. Both brain maps show the signatures' weights (positive = orange, negative = blue). For display purposes, the map of the SIIPS1 shows weights that exceed a predefined threshold (false discovery rate of $q < 0.05$). Intensity and unpleasantness ratings were given on a scale from 0 ("not at all painful/unpleasant") to 8 ("very painful/unpleasant"). NPS and SIIPS1 responses are plotted as standardized signature scores. Grey and red dots in violin plots represent single values and mean scores, respectively. * $< .05$, ** $< .01$, *** $< .001$, mark significant planned pairwise comparisons derived from the linear mixed models.

Reviewer #2 (Remarks on code availability):

I have not tried to run it, but the code (R code to run statistical analyses) seems useful.

Reviewer #3 (Remarks on code availability):

The provided code can be used in itself with the provided processed data. The plots and reported statistical test results from the manuscript main text can be reproduced.

The statistics and the corresponding retrospective rating's data in the supplementary are not provided.

Response to reviewer:

We appreciate the reviewer's valuable feedback and are happy that everything is sufficiently transparent. Initially, we intended to include only the data and scripts related to our primary analyses in the main text. However, following the revision process, we incorporated several additional analyses, which are now detailed in the Supplementary Information. In the spirit of open science, we have chosen to include all relevant data and analyses in our OSF repository. The scripts are organized into main and supplementary analyses (see lines 728-730 in main manuscript).

Reviewer #4 (Remarks on code availability):

I'm not the R user.

I do not think I'm the person who could not review the code properly. However, it seems that they provided the script for the analysis and the relevant data for the reproducibility.

Response letter for

Nature exposure induces analgesic effects by acting on nociception-related neural processing

General statement: We would like to sincerely thank the reviewers once again for their time and effort in carefully and thoughtfully evaluating our manuscript, especially within a short timeframe. We are pleased to have addressed all previous comments and concerns to the reviewers' satisfaction. Regarding the remaining minor comments and suggestions please find our detailed responses and accompanying changes in the manuscript below.

We would also like to express our gratitude to the handling editor for the expert guidance and efficiency in overseeing this manuscript through the review process.

We hope that the remaining minor revisions meet the expectations of both the reviewers and the editor.

REVIEWER COMMENTS

Reviewer #1 (Remarks to the Author):

The revised manuscript addresses all our comments and concerns. The authors have made commendable efforts to substantiate their conclusions, incorporating new analytical pipelines and additional analyses that significantly strengthen and validate their findings.

We have only two comments:

1. Reviewer: As suggested, the authors have now included correlation analyses between behavioral measures (pain ratings) and neural effects (ROI and NPS/SIIPs), supporting an association between these measures, as well-documented in the literature. However, to further substantiate the claim that the analgesic effect of nature is driven by changes in nociceptive processing, it would be beneficial for the authors to demonstrate correlations between the individual analgesic effect of nature (e.g., individual deltas in pain ratings) and the corresponding neural responses. I apologize if my earlier comments on this point were not sufficiently clear.

Response to reviewer:

We apologize, for not directly addressing the suggestion raised by the reviewers during the first round. Upon reexamination we realize the suggestion was sufficiently clear. In line with the reviewer's recommendation, we have now included correlations of delta (difference) scores between environments (for both immediate rating and neural responses) in the Supplementary Information. These analyses reveal the expected patterns. Larger differences between environments in self-reported pain correlate with larger differences in neural responses. However, it is important to note that the correlations are generally of low magnitude. However, given the constraints of low to moderate correlations between raw neural and subjective measurements in general (as reported in the last revision), we believe that the observed magnitude and pattern of correlations align with expectations. We have made the following amendments to the Supplement which we also briefly reference in the main text.

Main text changes

296-299

We also performed further exploratory analyses on the association between self-reported and neural pain responses (including the association of difference scores across environment pairs between immediate ratings and neural responses), as well as the role of immersion.

Supplementary Information text changes

623-655

Furthermore, in an exploratory additional analysis prompted by a reviewer comment, we calculated difference scores for immediate self-reported pain ratings and neural responses (NPS, SIIPS1, Thalamus, and pINS) between pairs of environments. Specifically, we computed the difference between the average response in the natural environment and the average response in the urban or indoor environments, separately for each dependent variable. Additionally, we calculated the difference between the average response in the urban and indoor environments. We then explored the correlations of these difference scores between self-reported pain ratings and neural responses for each environment pair (e.g., the correlation between the nature-urban difference in intensity ratings and Thalamus activity). For the nature-urban difference, we observed the following correlations: $r = .19$ ($p = .19$), $r = .23$ ($p = .11$), $r = .38$ ($p = .007$), $r = .29$ ($p = .043$), and $r = .19$ ($p = .183$) for intensity ratings, and $r = .27$ ($p = .064$), $r = .20$ ($p = .168$), $r = .43$ ($p = .002$), $r = .30$ ($p = .034$), and $r = .33$ ($p = .021$) for the unpleasantness ratings with the NPS, SIIPS1, Thalamus, S2, and pINS responses, respectively. For the nature-indoor difference, we observed similar correlations: $r = .19$ ($p = .19$), $r = .24$ ($p = .11$), $r = .15$ ($p = .319$), $r = .23$ ($p = .110$), and $r = .18$ ($p = .207$) for intensity ratings, and $r = .25$ ($p = .085$), $r = .19$ ($p = .182$), $r = .26$ ($p = .071$), $r = .40$ ($p = .004$), and $r = .33$ ($p = .018$) for the unpleasantness ratings with the NPS, SIIPS1, Thalamus, S2, and pINS responses, respectively. Finally, for the urban-indoor difference, we observed the following correlations: $r = .03$ ($p = .843$), $r = .04$ ($p = .805$), $r = .01$ ($p = .930$), $r = -.09$ ($p = .559$), and $r = -.13$ ($p = .371$) for intensity ratings, and $r = .14$ ($p = .336$), $r = -.11$ ($p = .452$), $r = .12$ ($p = .395$), $r = .13$ ($p = .386$), and $r = .13$ ($p = .361$) for the unpleasantness ratings with the NPS, SIIPS1, Thalamus, S2, and pINS responses,

respectively. Note that the correlations were not corrected for multiple comparisons due to the exploratory nature of this analysis. In general, we observed that larger differences in neural responses were associated with larger differences in self-reported ratings, with the most robust associations found for differences in nociception-related areas (i.e., Thalamus, S2). Notably, this was not substantiated by the associations observed between differences in self-reported ratings and the NPS or SIIPS1, which exhibited similarly modest correlations. Furthermore, the overall pattern revealed relatively low correlations, which is not surprising, given the inherent complexity of the measures and the well-documented finding that self-reported and neural pain data typically exhibit low to moderate associations, which was also confirmed by our study (see correlations reported above). Since these correlations accounted for 25% or less of the variance in neural responses (or vice versa) across environments in our data, the potential magnitude of the exploratory correlations of the differences across environments is inherently constrained.

2. Reviewer: The authors might also give an explanation why the provided MRI acquisition parameters of the fMRI sequence differ from the "classic" association between FOV, matrix size(MS), and voxel size(VS): $FOV=MS*VS$.

Response to reviewer:

We thank the reviewers for once again pointing out that this section of the manuscript was not sufficiently clear. Upon reflection, we now understand the source of confusion and miscommunication from our end. In the original manuscript, we provided the field of view (FOV) dimensions only for the specific directions corresponding to the acquired slices (i.e., sagittal for structural scans and axial for functional scans and field maps). We have now included all three FOV parameters. As shown, our parameters are consistent with the "classic" association and adhere to the formula: $FOV = MS * VS$.

Main text changes

624-636

Repetition time (TR) = 800 ms, echo time (TE) = 34 ms, flip angle = 50°, field of view (FOV) = 210 x 210 x 138 mm³, multi-band acceleration factor = 4, interleaved multi-slice mode, interleaved acquisition, matrix size = 96 x 96 x 36, voxel size = 2.18 x 2.18 x 3.84 mm³, 36 axial slices of the whole brain with slice thickness = 3.50 mm and an interslice gap of 0.34 mm. We used a magnetization-prepared rapid acquisition gradient echo sequence with the following parameters to obtain the structural image at the end of each scanning session: TR = 2,300 ms, TE = 2.29 ms, flip angle = 8°, FOV = 165 x 240 x 240 mm³, ascending acquisition, single shot multi-slice mode, 176 sagittal slices, matrix size = 176 x 256 x 256, voxel size = 0.94 x 0.935 x 0.935 mm³, slice thickness = 0.94 mm. Furthermore, field map images were acquired using a dual-echo gradient echo sequence to correct the functional images for magnetic field inhomogeneities, with the following parameters: TR = 400 ms, TE1 = 4.92 ms, TE2 = 7.38 ms, flip angle = 60°, FOV = 220 x 220 x 138 mm³, matrix size = 128 x 128 x 36, voxel size = 1.72 x 1.72 x 3.84 mm³, 36 axial slices aligned with the orientation of the functional images, and slice thickness = 3.84 mm.

Reviewer #2 (Remarks to the Author):

We appreciate your substantial efforts in revising the manuscript, including additional analyses of fMRI and behavioral data. We have only minor comments:

1. Reviewer: Some guidelines for interpreting the magnitude of effect sizes, e.g., d_{rm} , would be helpful.

Response to reviewer:

We appreciate the reviewer's suggestion, to include direct guidelines for interpreting the magnitude of effect sizes, as this facilitates interpretation for the readership. In response, we have now incorporated a concise reference to such guidelines in the main manuscript. Note that d_{rm} accounts for the repeated measures correlation and can therefore be interpreted using the classical benchmarks for Cohen's d from an independent group design.

Main text changes

714-716

We interpreted effect sizes based on widely used conventions⁵⁹, where small effects are defined as 0.2, medium effects as 0.5, and large effects as 0.8.

2. Reviewer: Clarification on how standardized signature scores were calculated in Figs. 2D and 2F would also be helpful.

Response to reviewer:

We thank the reviewer for emphasizing the importance of clarifying the standardization procedure to enhance transparency. In line with this suggestion, we have now included a description in the main text.

Main text changes

697-702

For the neural signatures, we used the **standardized signature response of the NPS and SIIPS1 as the dependent variable to be predicted by the fixed effect of environment (nature as a reference), signature (NPS as reference), and their interaction (with random slopes and intercepts for environment and signature by participant). Standardization across responses was performed separately for each signature, pooled across environments.**

3. Reviewer: The clusters reported in Supplementary Table 3 appear overly large, reducing their utility for identifying regions. It would be helpful if they could provide information about subclusters or peak information.

Response to reviewer:

We agree with the reviewer that the clusters are quite large. To facilitate interpretation, we have now included subclusters (voxel-wise corrected) for the two main clusters (cluster-wise corrected), as well as a more conservatively thresholded map of the activity in the Supporting Information. We hope that, in combination, these additions will allow the interested readership to more closely examine the well-established activity patterns associated with first-hand pain.

Supporting Information text changes:
746-746 & 697-707

Supplementary Table 3. Whole brain results for the contrast pain>no-pain across all environments.

Clusters and brain regions	h	MNI			t-value	p
		x	y	z		
Cluster 1 (k = 31099)	R	38	10	-2	10.43	<.000001
Insular Cortex (anterior)	L	-34	8	10	10.16	
Insular Cortex (posterior)	R	38	-16	12	10.13	
Cluster 2 (k = 13893)	R	12	-84	6	9.19	<.000001
Lingual Gyrus	L	-2	-80	-2	8.95	
Occipital Pole	R	8	-96	-14	8.92	

Note: Significant clusters resulting from the contrast [pain>no-pain]. h = hemisphere, k = cluster size, MNI coordinates x, y, z, t-value, and p-value (FWE-corrected at $p < .05$; cluster-level for Cluster 1 and 2; voxel-level for sub-clusters). Peak coordinates were labeled according to the automated anatomical labeling (AAL2) atlas²⁹.

Supplementary Fig. 1 Whole-brain results of pain>no-pain contrast across environments. Since the identified clusters at the a priori FWE-corrected threshold turned out to be rather large, we present the statistical activation maps at two thresholds to facilitate the identification and delineation of activated regions: **a** A priori threshold with FWE-correction at $p < .05$; **b** More conservative threshold with FWE-correction at $p < .001$. The glass brain shows activity changes in the hemodynamic response triggered by electrical stimulations of the left hand. We observed increased activity in a wide variety of brain regions associated with processing first-hand pain. L = left hemisphere, R = right hemisphere; color bar depicts t-values. Source data are provided as a Source Data file.